# How Much Data Are Augmentations Worth? An Investigation into Scaling Laws, Invariance, and Implicit Regularization

**Jonas Geiping**
University of Maryland, College Park

**Micah Goldblum**
New York University

**Gowthami Somepalli**
University of Maryland, College Park

**Ravid Shwartz-Ziv**
New York University

**Tom Goldstein**
University of Maryland, College Park

**Andrew Gordon-Wilson**
New York University

## Abstract

Despite the clear performance benefits of data augmentations, little is known about why they are so effective. In this paper, we disentangle several key mechanisms through which data augmentations operate. Establishing an *exchange rate* between augmented and additional real data, we find that in out-of-distribution testing scenarios, augmentations which yield samples that are diverse, but inconsistent with the data distribution can be even more valuable than additional training data. Moreover, we find that data augmentations which encourage invariances can be more valuable than invariance alone, especially on small and medium sized training sets. Following this observation, we show that augmentations induce additional stochasticity during training, effectively flattening the loss landscape.

## 1 Introduction

Even with the proliferation of large-scale image datasets, deep neural networks for computer vision represent highly flexible model families and often contain orders of magnitude more parameters than the size of their training sets. As a result, large models trained on limited datasets still have the capacity for improvement. To make up for this data shortage, standard operating procedure involves diversifying training data by augmenting samples with randomly applied transformations that preserve semantic content. These augmented samples expand the volume of data available for training, resulting in downstream performance benefits that one might expect from a larger dataset. However, the now profound significance of data augmentation (DA) for boosting performance suggests that its benefits may be more nuanced than previously believed.

In addition to adding extra samples, augmentation promotes invariance by encouraging models to make consistent predictions across augmented views of each sample. The need to incorporate invariances in neural networks has motivated the development of architectures that are explicitly constrained to be equivariant to transformations (Weiler & Cesa, 2019; Finzi et al., 2020). If the downstream effects of data augmentations were attributable solely to invariance, then we could replace DA with explicit model constraints. However, if explicit constraints cannot replicate the benefits of augmentation, then augmentations may affect training dynamics beyond imposing constraints.

Finally, augmentation may improve training by serving as an extra source of stochasticity. Under DA, randomization during training comes not only from randomly selecting samples from the dataset to form batches but also from sampling transformations with which to augment data (Fort et al., 2022). Stochastic optimization is associated with benefits in non-convex problems wherein the optimizer can bias parameters towards flatter minima (Jastrzębski et al., 2018; Geiping et al., 2021; Liu et al., 2021a).

In this paper, we re-examine the role of data augmentation. In particular, we quantify the effects of data augmentation in expanding available training data, promoting invariance, and acting as a source of stochasticity during training. In summary:

- We quantify the relationship between augmented views of training samples and extra data, evaluating the benefits of augmentations as the number of samples rises. We find that augmentations can confer comparable benefits to independently drawn samples on in-domain test sets and even stronger benefits on out-of-distribution testing.
- We observe that models that learn invariances via data augmentation provide additional regularization compared to invariant architectures and we show that invariances that are uncharacteristic of the data distribution still benefit performance.
- We then clarify the regularization benefits gained from augmentations through measurements of flatness and gradient noise showing how DA exhibits flatness-seeking behavior.

## 2 RELATED WORK

**Data Augmentations in Computer Vision.** Data augmentations have been a staple of deep learning, used to deform handwritten digits as early as Yaeger et al. (1996) and LeCun et al. (1998), or to improve oversampling on class-imbalanced datasets (Chawla et al., 2002). These early works hypothesize that data augmentations are necessary to prevent overfitting when training neural networks since they typically contain many more parameters than training data points (LeCun et al., 1998).

We restrict our study to augmentations which act on a single sample and do not modify labels. Namely, we study augmentations which can be written as $(T(\mathbf{x}), y)$, where $(\mathbf{x}, y)$ denotes an input-label pair, and $T \sim \mathcal{T}$ is a random transformation sampled from a distribution of such transformations. For a broad and thorough discussion on image augmentations, their categorization, and applications to computer vision, see Shorten & Khoshgoftaar (2019) and Xu et al. (2022). We consider basic geometric (random crops, flips, perspective) and photometric (jitter, blur, contrast) transformations, and common augmentation policies, such as AutoAug (Cubuk et al., 2019), RandAug (Cubuk et al., 2020), AugMix (Hendrycks et al., 2020) and TrivialAug (Müller & Hutter, 2021) which combine basic augmentations.

**Understanding the Role of Augmentation and Invariance.** Works such as Hernández-García & König (2018) propose that data augmentations (DA) induce implicit regularization. Empirical evaluations describe useful augmentations as "label preserving", namely they do not significantly change the conditional probability over labels (Taylor & Nitschke, 2018). Gontijo-Lopes et al. (2020b;a) investigate empirical notions of consistency and diversity, and variations in dataset scales (Steiner et al., 2022). They measure *consistency* (referred to as affinity) as the performance of models trained without augmentation on augmented validation sets. They also measure *diversity* as the ratio of training loss of a model trained with and without augmentations and conclude that strong data augmentations should be both consistent and diverse, an effect also seen in Kim et al. (2021). In contrast to Gontijo-Lopes et al. (2020b), Marcu & Prugel-Bennett (2022) find that the value of data augmentations cannot be measured by how much they deform the data distribution. Other work proposes to learn invariances parameterized as augmentations from the data (Benton et al., 2020), investigates the number of samples required to learn an invariance (Balestriero et al., 2022b), uncovers the tendency of augmentations to sacrifice performance on some classes in exchange for gains on others (Balestriero et al., 2022a), or argues that data augmentations cause models to misrepresent uncertainty (Kapoor et al., 2022).

Theoretical investigations in Chen et al. (2020a) formalize data augmentations as label-preserving group actions and discuss an inherent *invariance-variance* trade-off. Variance regularization also arises when modeling augmentations for kernel classifiers (Dao et al., 2019). For a binary classifier with finite VC dimension, the bound on expected risk can be reduced through additional data generated via augmentations until *inconsistency* between augmented and real data distributions overwhelms would-be gains (He et al., 2019b). The regularizing effect of data augmentations is investigated in LeJeune et al. (2019) who propose a model under which continuous augmentations increase the smoothness of neural network decision boundaries. Rajput et al. (2019) similarly find that linear classifiers trained with sufficient augmentations can approximate the maximum margin solution. Hanin & Sun (2021) relate data augmentations to stochastic optimization. A different angle towards understanding invariances through data augmentations is presented in Zhu et al. (2021), where the effect of DA in increasing the theoretical sample cover of the distribution is investigated, and augmentations can reduce the amount of data required, if they "cover" the real distribution.

**Stochastic Optimization and Neural Network Training.** The implicit regularization of SGD is regarded as an essential component for neural network generalization (An, 1996; Neyshabur et al., 2017). Stochastic training which randomizes gradients can drive parameters into flatter minima,

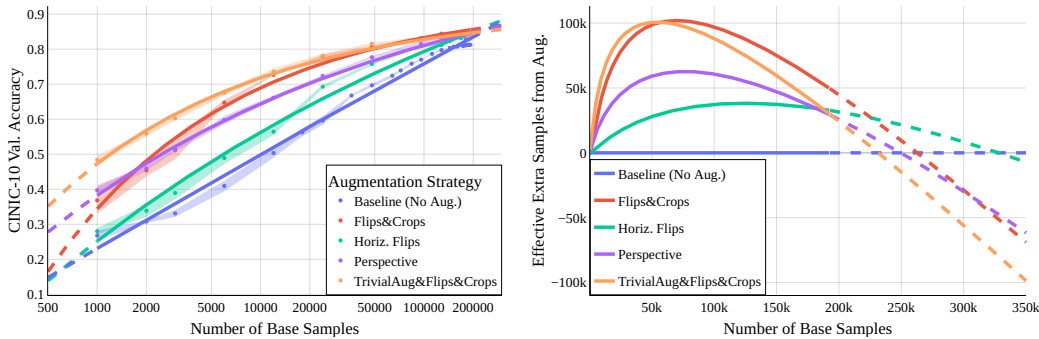

**Figure 1: Power laws** $f(x) = ax^{-c} + b$ for select augmentations applied randomly and the gain in terms of effective extra samples from Equation (1). Fitted curves marked in solid colors, with extrapolated regions dashed. **Left:** Number of base samples (from CINIC-10) on the logarithmic horizontal axis compared to validation accuracy. The scaling behavior of each augmentation is closely matched by these power laws. **Right**: Number of base samples compared to effective extra data, showing how the benefits of each data augmentation scale as the model is trained on more and more data. Policies that are strong but inconsistent, such as TrivialAug, reach the highest peak benefit (at 50 000 base samples TrivialAug generates effectively 100 000 extra samples), but also fall off faster than consistent augmentations, such as horizontal flips, which provide benefits up to 350 000 base samples.

associated with superior generalization (Jastrzębski et al., 2018; Huang et al., 2020; Liu et al., 2021a). In fact, Geiping et al. (2021) find that neural networks trained with non-stochastic full batch gradient descent require explicit flatness-seeking regularizers in order to achieve comparable test accuracy. Data augmentations provide an additional source of stochasticity during training on top of batch sampling, which we will investigate in this work.

We fuse together the above three topics and explore the role data augmentations holistically at scale.In doing so, we fill in several gaps in the literature discussed in this section. Unlike work which quantifies data augmentations in terms of accuracy boosts for fixed sample sizes, we compare the benefits of augmentations to those achieved by instead collecting more data. While other works have studied the role of data augmentations in learning invariance, we find that even invariances which have no relationship to invariances in the training data distribution are still effective. Finally, we develop an understanding of batch augmentation by showing that stochastically applied augmentations increase gradient noise during training, leading to qualitatively distinct minima.

## 3    AUGMENTATIONS AS ADDITIONAL DATA

A central role of data augmentation is to serve as *extra data* and expand limited datasets used for training large models. In this section, we quantify this property, conducting a series of experiments culminating in measurements *exchange rates*, which indicate exactly how much data an augmentation is worth – the number of additional data samples which would yield the same performance gain as the augmentation. Such exchange rates constitute a novel angle for quantifying the practical benefits of data augmentations and conceptualize qualitative notions of consistency, diversity, and robustness to distribution shifts, and allow us to observe the properties of data augmentations over a wider range of dataset sizes. We conduct these experiments on subsets of the CINIC-10 dataset (Darlow et al., 2018), a drop-in replacement for CIFAR-10 (Krizhevsky, 2009), which contains ∼200 000 samples. This allows us to train models with augmented data on subsets similar to CIFAR-10, but compare to reference models trained without augmentations on larger datasets. We start with ResNet-18 architectures, evaluating their exchange rate behavior, but consider other architectures in Section 3.1.2. Our dataset setup is quite specific to CIFAR-10/CINIC-10, revisiting this experiment e.g. for ImageNet would require running a pairing such as ImageNet/JFT-300 (Sun et al., 2017), which would be prohibitive for all experiments in this work. We include a validation of results on ImageNet in Appendix F and other datasets in Appendix B, to verify that the behaviors observed are not specific to CIFAR-10.

### 3.1    EXCHANGE RATES: HOW MANY SAMPLES ARE AUGMENTATIONS WORTH?

We visualize the validation accuracies for a range of models trained with select augmentations in Figure 1 (left). The validation behavior of these models can be well described by *power laws* of the form $f(x) = ax^{-c} + b$, describing the relationship between number of samples and validation

accuracy. From these power laws, we derive the exchange rates of various augmentations compared to the reference curve of un-augmented models $f_{\text{ref}} = a'x^{-c'} + b'$. For an augmentation policy described by $f_{\text{aug}} = ax^{-c} + b$, we define its exchange rate via

$$v_{\text{Effective Extra Samples from Augmentations}}(x) = f_{\text{ref}}^{-1}\left(f_{\text{aug}}(x)\right) - x \tag{1}$$

for a base dataset size $x$. We visualize this quantity in Figure 1 (right). This metric measures exactly the gain in accuracy between the un-augmented and augmented power laws shown on the left-hand side and converts this quantity into additional samples by evaluating on how much *more data* the reference models would need to realize the same accuracy as the augmented model.

A core property of augmentations that becomes evident in this analysis is the relationship between *consistency* of augmented samples with the underlying data distribution (Taylor & Nitschke, 2018; Gontijo-Lopes et al., 2020b; He et al., 2019b) and *diversity* of extra data (Gontijo-Lopes et al., 2020b). In Figure 1, we find that an augmentation strategy, such as TrivialAug (Müller & Hutter, 2021) (a policy consisting of a random draw from a table of 14 common photo- and geometric transforms, applied in tandem with horizontal flips and random crops), shown in orange, is highly diverse, generating a large amount of effective extra samples when the number of base samples is small. However, this policy also falls off quickest as more base samples are added: The policy is ultimately *inconsistent* with the underlying data distributions and when enough real samples are present, gains through augmentation deteriorate. On the flip side, augmenting with only horizontal flips, shown in green, is less diverse, and hence yields limited impact for smaller dataset sizes. However, the augmentation is much more consistent with the data distribution, and as such horizontal flips are beneficial even for large dataset sizes.

> **Takeaway**: The impact of augmentations is linked to dataset size; diverse but inconsistent augmentations provide large gains for smaller sizes but are hindrances at scale. Estimation of the value of future data collection should take this effect of augmentations on scaling into account.

### 3.1.1 MEASURING DIVERSITY AND CONSISTENCY DIRECTLY

We can disambiguate the effects of consistency and diversity further by analyzing these augmentations for a moment not as randomly applied augmentations, but as fixed enlargements of the dataset, *replacing* each base sample with a *fixed number of augmented views*. In Figure 2 (right), we see that a replacement of each sample with a single augmented view generated by TrivialAug is detrimental to performance (see single repetition plotted in red). As such, TrivialAug is *inconsistent* with the data distribution. On the other hand, evaluating the gains realized by replicating each sample in the dataset a fixed number of times via the augmentation policy, we also see that TrivialAug leads to significantly more diversity in this controlled experiment, compared to random flips (Figure 2, left).

Another way to look at Figure 2 is again as a measurement of extra data. Simply replacing each base sample by a single augmented sample is not a benefit (see single repetition plotted in red), yet we quickly find that large gains can be attributed simply to the duplication (green) and quadruplication (purple) of existing data and fixed multiplication can be worth substantial extra data.

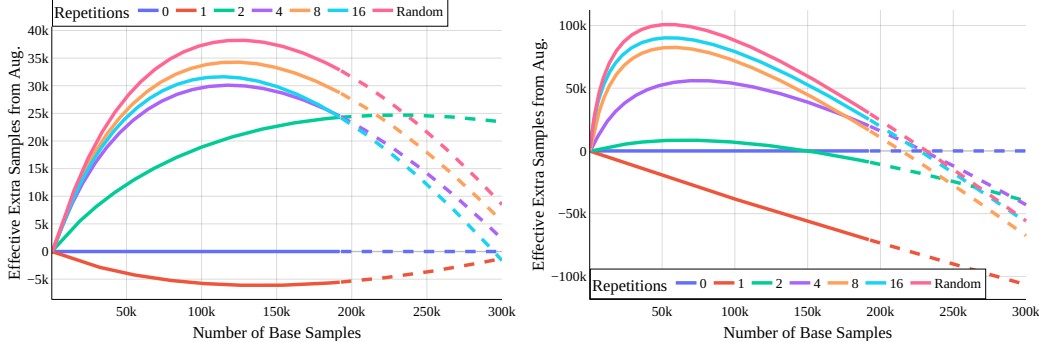

**Figure 2:** Exchange Rates via Equation (1) when creating **larger datasets** from a *fixed* number of repetitions of base samples via random data augmentations. Zero repetitions denote the unaugmented data. **Left:** Exchange rate for random horizontal flips as the augmentation policy is repeated. **Right:** Exchange Rates for TrivialAug. Even a few repetitions of existing samples generate most of the effective additional data observed in Figure 1.

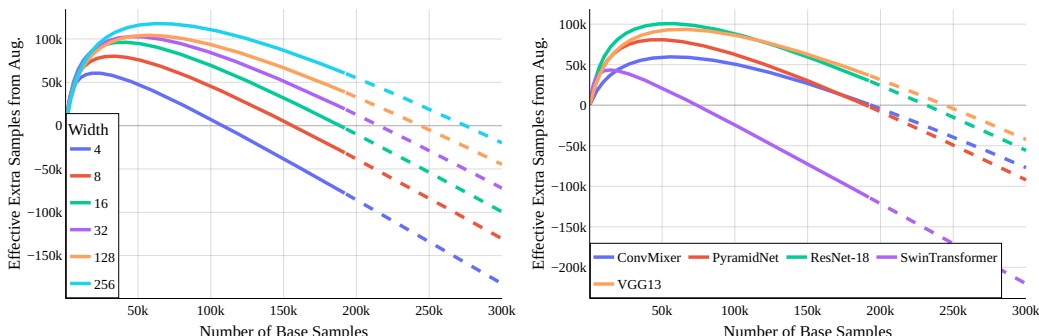

**Figure 3:** Exchange Rates via Equation (1) for *TrivialAug* when modifying the **model architecture**. Rates for various **widths of a ResNet-18 (left)**. 64 is the default in other parts of this work. Exchange Rates for select **vision architectures (right)** with similar parameter counts. Exchange rates behave similarly for a range of models architectures, but quantitative benefits increase as models widen and capacity increases.

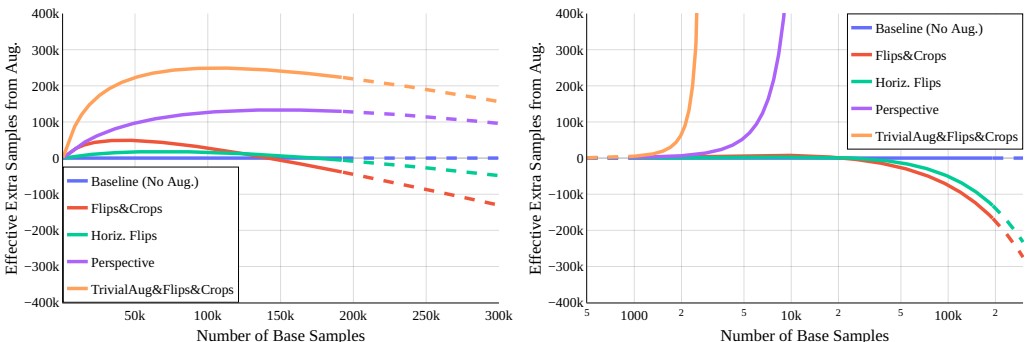

**Figure 4: Exchange Rates** as Figure 1, but evaluated on **CIFAR-10 validation (left)** and **CIFAR-10-C (right)**. Note that for CIFAR-10-C, inconsistent augmentations are worth more than any number of additional in-domain samples. Augmentations really are worth much more, under slight (left) and large (right) distribution shifts.

### 3.1.2 Exchange Rates for Model Scaling and Model Variations

Figure 3 evaluates the effective samples gained from augmentation as model width increases (left) and model architecture changes (right). For both plots, the references $f_{ref}$ are based on the unaugmented models with that same configuration of width or architecture. We find that model width of the evaluated ResNet-18 reliably correlates with the gains from augmentations to larger dataset sizes. Evaluating different architectures, we find that while global behavior is similar for all models, the exact exchange rates are similar for the convolutional architectures of ResNet, PyramidNet (Han et al., 2017) and VGG (Simonyan & Zisserman, 2014), mirroring their closely related inductive biases. On the other hand, from the two transformer architectures, we find that a notably limited benefit from augmentations versus real data for the Swin-Transformer (Liu et al., 2021b; 2022), but a large benefit for the ConvMixer (Trockman & Kolter, 2022). In direct comparison of both architectures, the Swin-Transformer contains a large number of features designed specifically for vision tasks, whereas ConvMixer is much more general, so that their differing inductive biases are again reflected in Figure 3.

> **Takeaway**: Relative gains through augmentations as sample sizes scale are broadly consistent across model widths and architectures, and absolute gains increase with model capacity.

### 3.2 Exchange Rates in Out-of-Distribution Settings

A benefit of extra data generated from augmentations that is often underappreciated is illustrated in Figure 4, showing exchange rates of the same models trained on CINIC-10 as analyzed in Figure 1, but evaluated on the CIFAR-10 validation set (left). Comparing CINIC-10 and CIFAR-10, both datasets are nearly indistinguishable using simple summary statistics (Darlow et al., 2018), yet there is a minor distribution shift caused by different image processing protocols during dataset curation. In this setting,

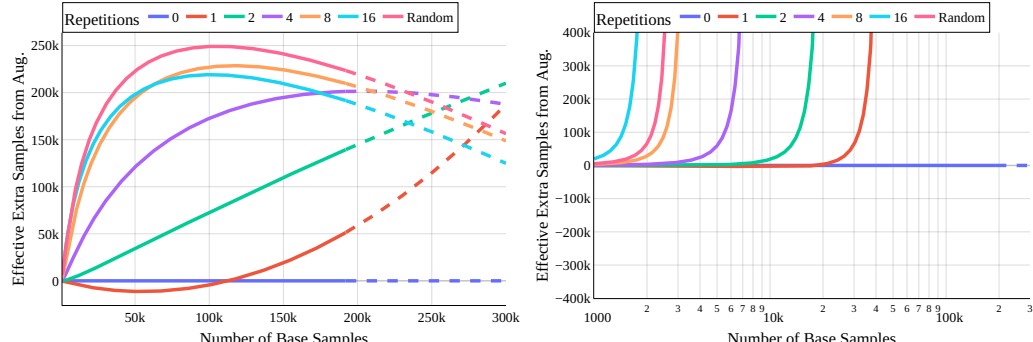

**Figure 5: Exchange Rates** from a *fixed* number of repetitions of samples via *TrivialAug*. "0" repetitions denote the unaugmented data. As Figure 2, but evaluated on **CIFAR-10 val. (left)** and **CIFAR-10-C (right)**. We find that even for this mild OOD shift, replacing each sample via augmentation (red) wins above 75 000 samples. For CIFAR-10-C, even a single repetition is worth more than any almost number of additional in-domain samples.

diverse augmentations are now quickly on-par with models trained on many more samples. Figure 5 shows that this effect is apparent, even if base samples are replaced by a fixed number of augmented samples. Four augmented samples are quickly worth more than additional real samples, and with enough base samples, even replacing each sample by a fixed augmented version is beneficial. These effects can be exaggerated by evaluating on the CIFAR-10-C dataset of common corruptions applied to CIFAR-10 (Hendrycks & Dietterich, 2018). There, we quickly find that evaluating exchange rates for this large distribution shift, that the stronger augmentations evaluated, quickly produces benefits beyond what a collection containing even substantial amounts of in-domain data would achieve.

The increased diversity in these augmentations broadens the support of the data distribution, and the support of the CIFAR-10 dataset appears to be well contained within the transformed data. In practical applications in domain adaptation (Zhang et al., 2019; Ahuja et al., 2021) and unsupervised learning (Chen et al., 2020b), actively broadening the support of the data distribution in the face of uncertainty is advantageous as suddenly each augmentation is effectively worth much more data.

> **Takeaway**: Data augmentations broaden the support of training data, which significantly extends their usefulness even on larger dataset scales in OOD testing scenarios.

## 4 AUGMENTATIONS AND INVARIANCE

The success of augmentations is often attributed to the invariances encoded into the model by enforcing the assignment of identical labels across transformations of each training sample. If the success of data augmentations can be attributed solely to invariance, then we can build *exactly invariant* models that achieve comparable accuracy when trained without data augmentation. Several works propose such mechanisms for constraining neural network layers to be invariant, and we will leverage these in our study.

### 4.1 INVARIANT NEURAL NETWORKS WITHOUT AUGMENTATION

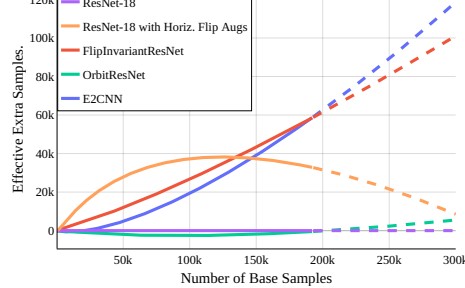

**Figure 6:** Exchange Rates for horiz. flips of ResNet and various invariant architectures. All models have an equal number of base parameters and are based on a Resnet-18 template. Invariant architectures show opposite scaling behavior to augmentations.

In order to probe the benefits of invariance without augmentations, we evaluate the following three methods for constructing invariant networks, for the case of invariance to horizontal flips:
**Prediction averaging:** Insert augmented views of a sample into the model and average the corresponding predictions (Simonyan & Zisserman, 2014). We use this procedure during both training and inference , and refer to the approach applied to a ResNet as flip-invariant ResNet-18.
**Orbit Selection:** An invariance can also be enforced via orbit selection (Gandikota et al., 2022),

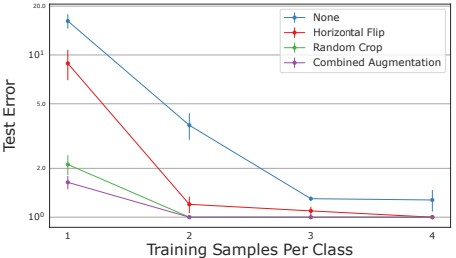 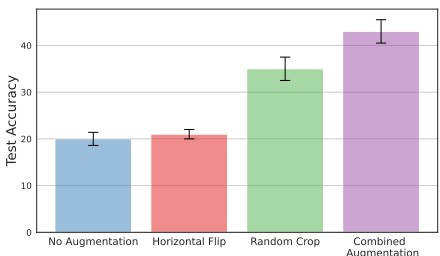

**Figure 7: Out-of-distribution augmentations still boost performance. Left:** Test error (log-scale) as a function of training samples when test images are rotations of training images. **Right:** Test accuracy on rotated samples from CIFAR-10 that were not used for training. All experiments performed on rotated CIFAR-10 samples with the ResNet-18 architecture. Bars mark standard error over 5 trials.

which corresponds to a normalization of the invariance. Here, an orbit mapping uniquely selects a single element from the group of transformation before the data is passed into the model.

**E2CNN:** General E(2)-Equivariant Steerable CNN (E2CNN) (Weiler & Cesa, 2019) constrains convolutional kernels to reflect a group equivariance. We instantiate an E2CNN with the same architecture as the ResNet-18 studied so far.

In Section 3, we saw that horizontal flips are consistent with the CIFAR-10 distribution, so it may not be surprising that horizontal flip invariant networks perform better than those trained with neither augmentations nor invariance constraints. Moreover, networks trained with data augmentations outperform invariant models for small and medium sample sizes. However, we observe in Figure 6 that all investigated invariant architectures (red, green, blue) catch up to models trained with random augmentations (in orange) as we increase the number of base samples. We will see in Section 5 that data augmentations serve as flatness-seeking regularizers, and the benefits of this additional regularization wane as the number of samples increases.

### 4.2 OUT-OF-DISTRIBUTION AUGMENTATIONS STILL IMPROVE PERFORMANCE

Previously, we observed the performance benefits of data augmentations which promote invariances consistent with the data distribution, or approximately so. But can it still be useful to augment our data with a transformation that generates samples completely outside the support of the data distribution and which are inconsistent with any label? To answer this question, we construct a synthetic dataset in which the exact invariances are known.

We begin by randomly sampling a single base image from each CIFAR-10 class. We then construct 10 classes by continuously rotating each of the base images, so that all samples in a class correspond to rotations of a single image. Thus, the classification task at hand is to determine which base image was rotated to form the test sample. We randomly sample rotations from each of these classes to serve as training data and another disjoint set of rotations to serve as test data. We then use horizontal flip and random crop data augmentations to generate out-of-distribution samples, since horizontally flipped or cropped image views cannot be formed merely via rotation. Note that this experiment is distinct from typical co-variate shift setups where the distribution of data domains differs, but the support is far from disjoint and may even be identical.

In Figure 7, we see that these out-of-distribution augmentations are beneficial nonetheless. Notably, random crops, which can generate significantly more unique views than horizontal flips, yield massive performance boosts for identifying rotated images, even though we know that the cropped samples are out-of-distribution. We also see in this figure that random crops are especially useful if we instead use as our test set rotations of samples from CIFAR-10 that were not used for training. Specifically, we assign a base test image and its rotations the same label as the base image from the training set with the same CIFAR-10 label. This experiment supports the observations from Section 3 that augmentations can be particularly beneficial for OOD generalization.

> **Takeaway**: Comparing invariant architectures to augmentations, we find that augmentations dominate on smaller scales, but invariant architectures catch up in the large-sample regime. Augmentations can provide benefits even on apparently unrelated invariances, which is particularly helpful for OOD generalization.

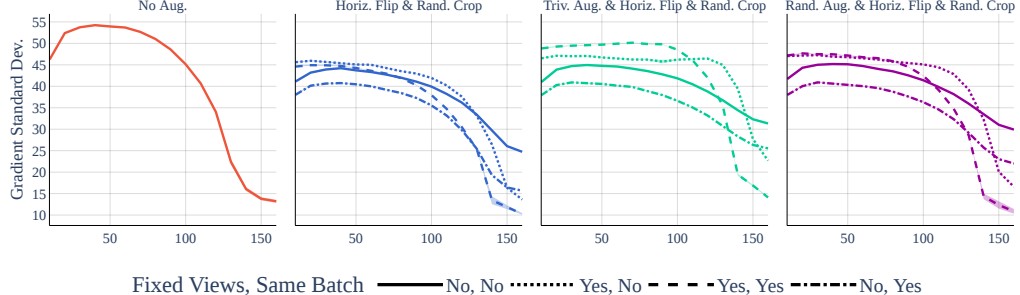

**Figure 8: Randomly applied augmentations significantly increase stochasticity late in training but decrease stochasticity early.** Standard deviation of gradient across epochs for different augmentations and different mini-batch sampling strategies. Shown is the mean over 10 runs, and shaded regions represent one standard error.

## 5 AUGMENTATIONS AS A SOURCE OF STOCHASTICITY DURING TRAINING

Typical loss functions are summed over training samples. During optimization, gradients are computed using small mini-batches of random samples, resulting in stochasticity. Augmentations increase the number of available data, often so much that we never sample the same data twice.

Since data augmentations expand and diversify the training set, they may serve as additional sources of stochasticity during optimization. If data augmentations do increase the variety of gradients, they could as a result cause us to find qualitatively different minima. Stochastic optimization is thought to be associated with flat minima of the loss landscape which are in turn associated with superior generalization (Jastrzębski et al., 2018; Huang et al., 2020; Liu et al., 2021a). This *flatness-seeking behavior* may be the effect of both the augmented loss function and also how we sample it.

To put this hypothesis to the test, we measure the standard deviation of gradients during optimization for models trained with and without data augmentations, and we quantify the flatness of the corresponding minima. We construct experiments that disentangle the augmented loss function from the additional stochasticity produced by sampling augmentations. We consider a "same batch" strategy in which gradient updates are averaged over multiple views of a single image, resulting in lower stochasticity. We also consider "fixed views" where we repeat a frozen set of augmented views per element of the training set, as in Figure 2.

### 5.1 MEASURING STOCHASTICITY

To measure stochasticity, we train a model on a given training set and augmentation strategy, and we freeze the model every 10 epochs to estimate the standard deviation (formally the norm of parameter-wise standard deviations) of its gradients over randomly sampled batches comprising 128 base images, the same batch size used during training. That is, we measure the square root of the average squared distance between a randomly sampled batch gradient and the mean gradient. We adopt a filter-normalized distance function (Li et al., 2018; Huang et al., 2020) to account for invariances in neural networks whereby shrinking the parameters in convolutional filters may not effect the network's output but may make the model more sensitive to parameter perturbations of a fixed size.

In Figure 8, we see that non-augmented datasets actually yield noisier gradients early in training, but this noise vanishes rapidly as over-fitting occurs. In contrast, randomly applied data augmentations result in flatter curves, indicating that the added diversity of views available for sampling preserves stochasticity later in training. We also see that for each augmentation policy, applying augmentations randomly results in the most stochasticity late in training, while including multiple random views in the same batch (Hoffer et al., 2020; Fort et al., 2022) results in less. Sampling augmentations from a fixed set of four views per sample (denoted "fixed views") results in even less stochasticity, and including each of the four views in every batch results in the least stochasticity (denoted "fixed vews", "same batch"). This ordering, which holds across all data augmentations we try, is consistent with the intuition that more randomness in augmentation leads to more stochasticity in training, notably only manifesting during later epochs. We will now see that the late-training stochasticity we measure correlates strongly with the flatness of the minima these optimization procedures find.

## 5.2 MEASURING FLATNESS

We adopt the flatness measurements from Huang et al. (2020) as these measurements are non-local, do not require Hessian computations which are dubious for non-smooth ReLU networks, and they are consistent with our filter-normalized gradient standard deviation measurements. Specifically, we measure the average filter-normalized distance in random directions from the trained model parameters before we reach a loss function value of 1.0, where loss is evaluated on the non-augmented dataset. Under this metric, larger values correspond to flatter minima as parameters can be perturbed further without increasing loss. We use the same ResNet-18 models trained in the stochasticity experiments above with the same exact augmentation setups.

Investigating Figure 8 and Figure 9, (see also Table 4), we observe that flatness correlates strongly with late-training stochasticity. Models trained without augmentation or with non-random augmentation, where all views are seen in each batch, are less stochastic at the end of training and find sharper minima. While previous works have associated SGD with flatness-seeking behavior (Jastrzębski et al., 2018; Geiping et al., 2021), data augmentations appear to contribute to this phenomenon. Simply put, training with randomized data augmentations finds flatter minima, and models trained with strong data augmentations lie at especially flat minima.

> **Takeaway**: Randomly applied augmentations provide benefits beyond invariance by flattening the loss landscape, which is reflected in both measurements of flatness after training and measurements of gradient noise late in the training.

## 5.3 DATASET SCALING AND FLATNESS

Figure 9 directly measures flatness for several dataset scales. We first notice that base models become flatter (with respect to their base samples) as the number of samples increases. Surprisingly, stronger augmentations can produce this effect quicker and raise flatness values even for lower sample sizes. As a notable example, TrivialAug produces models that remain relatively flat for all sample sizes considered. Furthermore, all augmentations converge to similar flatness in the sample size limit, as regularization becomes less relevant in the large data regime. Over all plots we can even correlate the

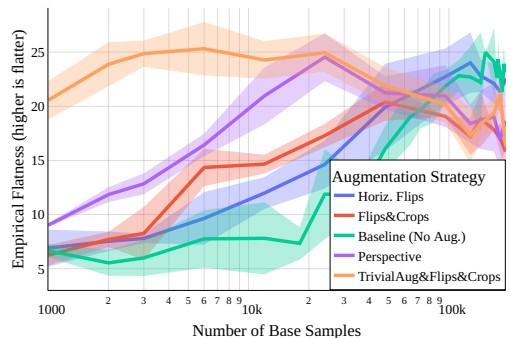

**Figure 9: Left:** Flatness for augmented models trained on several dataset sizes from Figure 1. All strategies converge to similar levels of flatness when scaling.

number of samples gained through augmentations and flatness and find that for all weaker augmentations, flatness of the solution is strongly correlated with the number of extra samples that are gained from the augmentation. We include more details in Appendix H.

> **Takeaway**: Strong data augmentations flatten the loss landscape to levels otherwise only reached with significantly larger datasets.

## 6 CONCLUSION

Data augmentations have a profound impact on the performance of neural networks, but their precise role has not been well understood; for example, if augmentations are simply a heuristic for learning certain symmetries, should we not prefer to directly encode these symmetries through advances in group equivariant networks? Through the lens of exchange rates and power laws, we observe the gains through augmentations as datasets scale and domains change. We find that augmentations dominating invariant architectures on smaller scales, but, scale in opposite ways. Augmentations are further distinguished from invariances in the way they can improve performance even out-of-distribution. Ultimately we find that we can connect these findings to the regularization effect induced by data augmentations, which we also measure, showing how augmentations flatten the loss landscape. This work promotes an all-encompassing understanding of neural network training, shedding light on the nuanced but significant role of data augmentation in the success of deep learning.

ETHICS STATEMENT

We foresee no direct negative societal consequences from this work. We do think that data augmentations are beneficial, especially in applications with only limited data, or where data curation is expensive. We argue that knowing how to exchange a smaller (but verified and curated) dataset for a larger dataset that is not augmented, but also due to its size less curated is hopefully helpful to the community.

REPRODUCIBILITY STATEMENT

We use an academic cluster with `NVIDIA RTXA4000` cards and also `NVIDIA GTX2080ti` cards. Each job is scheduled on a single GPU and the default setting of 60000 gradient steps takes roughly an hour to train and evaluate. Including all preliminary experiments we estimate a total usage of about 400 GPU days for this project. To replicate all experiments in the main body without repeated trials, we estimate a requirement of about 10 GPU days. We provide code at https://github.com/JonasGeiping/dataaugs and with the supplementary material.

ACKNOWLEDGEMENTS

This research is supported by NSF CAREER IIS-2145492, NSF I-DISRE 193471, NIH R01DA048764-01A1, NSF IIS-1910266, NSF 1922658 NRT-HDR, Meta Core Data Science, Google AI Research, BigHat Biosciences, Capital One, and an Amazon Research Award.

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

## A   EXPERIMENTAL SETUP

For all sections if not otherwise mentioned, we run the following protocol. We train the model (in the main body a ResNet-18), with stochastic gradient descent for 60000 steps with a batch size of 128. This corresponds to 160 epochs for a dataset of size 48000. We keep this number of gradient steps fixed when increasing or decreasing the dataset size. We linearly warm up the learning rate for

the first 2000 steps (about 5 epochs) up to peak rate of $0.1$ and then decay to zero by a half-cycle of cosine annealing. For all experiments, we include a standard weight decay of $5e-4$ and train with Nesterov momentum of $0.9$. The data is shuffled randomly after every epoch and we record validation accuracy every 10000 steps. All training runs are non-deterministic based on stochasticity due to random shuffling and `cudnn` non-determinism. We run at least five trials for each experiment in the main body and three trials for each in the supplementary material. In each plot, the standard deviation is shaded. For five trials this corresponds close to a $97.5\%$ confidence interval.

We use CIFAR-10 in its default configuration. For CINIC-10, we clean and resample the train and test sets. We first remove all CIFAR-10 train and test images from the dataset, we then further remove all exactly duplicated images and missing images, merge all remaining images and sample a new validation set of 10000 images. We provide code to replicate the creation of this cleaned dataset with the supplementary material. Overall we recover a new training set of size 193523. For CIFAR-10-C experiments in the supp. material, we report average accuracy over all transformations in CIFAR-10-C with a severity of 3. For all experiments where we consider only a subset of the existing data (e.g. each experiment with less than 193523 samples for CINIC-10), we sample a new subset of the training set for each experiment separately, to rule out confounding effects of good or bad splits of the training data, especially for smaller subset sizes. This new test set for CINIC-10 turn out to be harder than the CIFAR-10 test set, but we verify that the hyperparameters discussed above would result in more than 95% accuracy when training with CIFAR training data.

For experiments in the main body where data augmentations are randomly sampled a finite number of times, we store all augmentations in a database (`lmdb`) that is recreated in each run. As a result, each experiment contains a fixed set of finite views of each original datapoint, but these views are randomized across experiments. Due to random shuffling, samples from this enlarged dataset are drawn randomly and multiple views of the image are only guaranteed to occur in the same batch in the batch augmentation experiments in Section 5.

To compute measurements of exchange rates, we first compute the mean validation accuracy CINIC-10 for each experiment. We then train the reference models for CINIC-10 subset sizes of $1000, 2000, 3000, 6000, 12000, 24000, 48000, 96000, 128000, 144000, 168000, 180000, 192000$.

To estimate parameters $a, b, c$ for $f_p(x) = ax^{-c} + b$ in the exchange rate plots in all sections we use a non-linear least-squares algorithm, initialized from starting parameters that describe the curve for no augmentations. For this we use the Levenberg-Marquardt implementation of `MINPACK`, as wrapped in `scipy`.

To cross-reference the average validation accuracy of these reference models with our data augmentation experiments in Table 1, we assert that validation accuracies are monotonically increasing as subset sizes increase and fit a linear spline $f_{\text{ref}}$ for interpolation. We then compute the exchange ratios of Table 1 via $f_{\text{ref}}^{-1}(x)/b$, for the base dataset size $b$ which is 48000 in Table 1 and input mean validation accuracy $x$ for each experiment. For values outside the interval spanned by the minimal and maximal validation accuracies of the reference data, we reuse the power laws of the form $f_p(x) = ax^{-c} + b$ described in Sec. 3.3 and again compute $f_p^{-1}(x)/b$. We mark these extrapolated values by a $*$ in the table.

The ResNet-18 model employed in the model is a modern variant (He et al., 2016; 2019a) and contains the usual CIFAR-10 stem consisting of a single $3 \times 3$ convolutional layer without pooling, instead of the ImageNet stem (of two convolutional layers with stride and max-pooling). For experiments in the supplementary material, we further consider a ResNet-8 (i.e. three stages and a single block per stage) (He et al., 2016), a VGG-11 (Simonyan & Zisserman, 2014) with batch normalization, and a ConvMixer architecture (Trockman & Kolter, 2022) of depth 8 with hidden dimension 128 and spatial kernel size of 7.

We implement and run all experiments in `PyTorch` and make use of `torchvision` implementations for a range of data augmentations investigated in this work. We provide code to replicate all experiments at https://github.com/JonasGeiping/dataaugs and with the supplementary material.

## A.1 HYPERPARAMETERS FOR AUGMENTATIONS

For each augmentation we broadly follow established defaults. For completion we record these, and additional details here.

**Horiz. Flips:** A data point is flipped horizontally with probability 0.5.

**Det. Horiz. Flips:** Deterministic horizontal flips. For $1\times$, this corresponds to flipping every data point. $2\times$ corresponds to both flips being contained in the dataset.

**Vert. Flips:** A data point is flipped vertically with probability 0.5.

**Det. Vert. Flips:** Deterministic vertical flips. For $1\times$, this corresponds to flipping every data point. $2\times$ corresponds to both flips being contained in the dataset.

**Random Crops:** The image is padded by with zero-padding by 4 pixels in each direction and then a image of size $32 \times 32$ is cropped (This is classical random cropping for CIFAR-10).

**Flips&Crops:** Both random crops and horizontal flips are employed, as described above.

**Perspectives:** Performs a random perspective transform with probability 0.5 with bilinear resampling.

**Jitter:** Color jitter, randomly transforming contrast, hue and brightness of the image. For each distortion, sample a new scale uniformly from $[0.5, 1.5]$.

**Blur:** Blurs the image with a Gaussian blur with $\sigma = 3$.

**AutoAug:** Employ the augmentation policy of Cubuk et al. (2019), with the CIFAR-10 policy.

**AugMix:** The augmentation policy of Hendrycks et al. (2020).

**RandAug:** The augmentation policy of Cubuk et al. (2020), again with the CIFAR-10 policy.

**TrivialAug:** The augmentation policy of Müller & Hutter (2021) in its "wide" configuration.

**AutoAug&Flips&Crops:** The AutoAug policy followed by random horizontal flips and random crops as described above.

**AugMix&Flips&Crops:** The Augmix policy followed by random horizontal flips and random crops as described above.

**RandAug&Flips&Crops:** The RandAug policy followed by random horizontal flips and random crops as described above.

**TrivialAug&Flips&Crops:** The TrivialAug policy followed by random horizontal flips and random crops as described above.

## A.2 DATA LICENSING

We investigate, MNIST (LeCun et al., 1998), CIFAR-10 and CIFAR-100 (Krizhevsky, 2009), EMNIST (Cohen et al., 2017), CIFAR10-C (Hendrycks & Dietterich, 2018) and CINIC-10 (Darlow et al., 2018) and refer to these publications for additional details. We remove duplicates and missing data from CINIC-10 as described in the experimental setup.

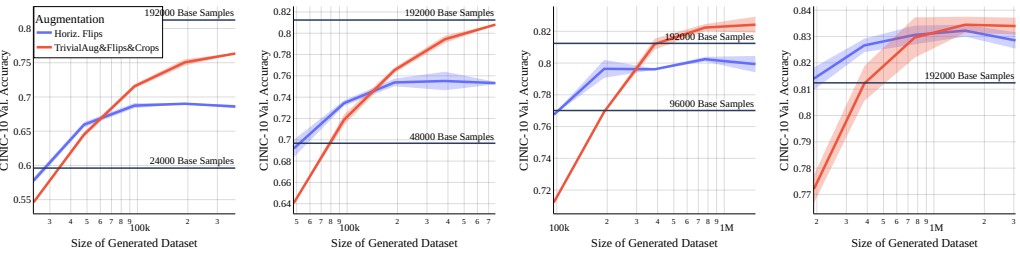

**Figure 10:** Repeated Data for CINIC-10. These are vertical slices through Figure 2. For (left-to-right): 24000, 48000, 96000, 192000 base samples.

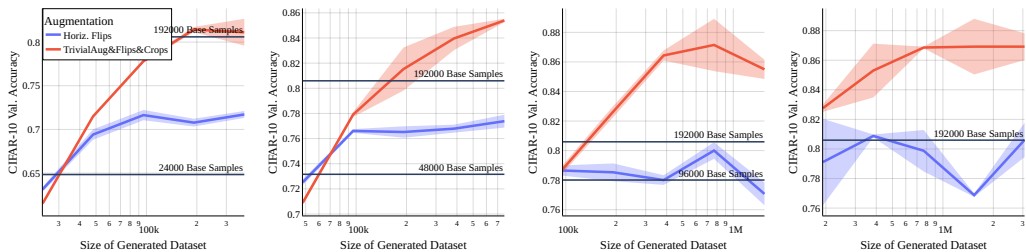

**Figure 11:** Repeated Data for CINIC-10, evaluated on CIFAR-10. These are vertical slices through Figure 5 (left). For (left-to-right): 24000, 48000, 96000, 192000 base samples.

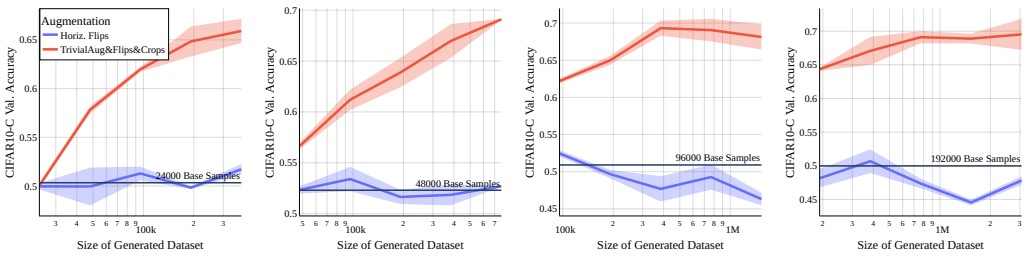

**Figure 12:** Repeated Data for CINIC-10, evaluated on CIFAR-10-C. These are vertical slices through Figure 5 (right). For (left-to-right): 24000, 48000, 96000, 192000 base samples.

## B ADDITIONAL EXPERIMENTAL RESULTS

We include additional material for the experimental section in a series of figures and tables. Table 1 is an extended table of the exchange rates for CINIC-10 for a sample size of 48000, including repetitions up to $32\times$ and ablating the number of steps (This is a slice of Figure 1). Behavior is consistent over additional repetitions, so we chose not to include these additional rows in the main body. This table further includes additional data augmentations not featured in the main body, over which behavior is consistent. Table 2 and Table 3 are then variants of this table where validation accuracy is evaluated on CIFAR-10 and CIFAR-10-C, respectively. CIFAR-10-C is a significant distribution shift that cannot be mitigated by additional CINIC-10 data, only training on, e.g. blurred samples, provides robustness to this distortion. We further find that training with horizontal flips in our experimental setup is quickly disadvantageous.

To provide additional clarification for Figure 2, we also provide slices through this plot at sample sizes of 24000, 48000, 96000 and 192000 in Figure 10 for CINIC validation data, Figure 11 for CIFAR-10 and Figure 12 for CIFAR-10-C data. Further, Figure 13 shows the data points underlying Figure 3 for additional clarification.

An additional table comparing the end-of-training stochasticity in Figure 8 and flatness measurements in Figure 9 for ease of references is Table 4.

## C ADDITIONAL DATASETS AND MODELS

We further verify that the findings discussed in the main body are not limited to the choice of dataset and model therein and provide additional reproduction aside from the investigation into model architectures in Figure 3. We repeat the tabular representation of exchange rates and a simplfied form of Figure 11 for a tiny ResNet-8 in Figure 14 and Table 5, a VGG-11 in Figure 15 and Table 6 and a ConvMixer (as representative of modern ConvNet/Transformer variations) in Figure 16 and Table 7.

We then further repeat these experiments with models trained on the MNIST training set in Figure 17 and Table 8, CIFAR-100 in Figure 18 and Table 9, as well as EMNIST in Figure 19 and Table 10.

We further include repeated experiments for Sec. 5 on CIFAR-100 in Figure 20 and Table 11.

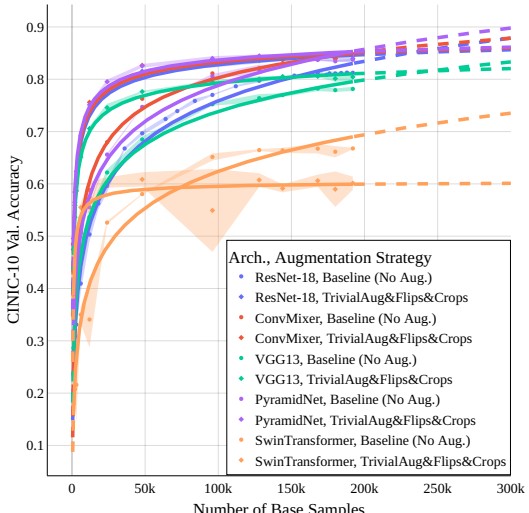

**Figure 13: Power laws** $f(x) = ax^{-c} + b$ for TrivialAug with flips and crops applied randomly. Equation (1). Fitted power law curves marked in solid colors, with extrapolated regions dashed. Standard deviation around measured samples (dotted) is shaded. Figure 3 is based on this data, comparing exchanges from unaugmented to augmented separately for each model.

**Table 1:** Extended table of **Exchange rates** for augmentations applied to 48000 base samples from the CINIC-10 training set, compared to reference models trained without augmentations on up to 192000 samples. We measure the exchange rate w.r.t. accuracy on the in-domain **CINIC-10 val. set**. Values marked with $*$ fall outside the range of reference datasets and are extrapolated using power laws. For a select augmentations we also include experiments with 240000 steps, i.e. 640 passes through the data to verify the utility of our chosen schedule of 60000 steps.

| Augmentation | CINIC-10 (in-domain) | | | | | | | |
| --- | --- | --- | --- | --- | --- | --- | --- | --- |
| | 1x | 2x | 4x | 8x | 16x | 32x | rand (160) | rand (640) |
| - | 1.00 | 1.00 | 1.00 | 1.00 | 1.00 | 1.00 | 1.00 | 1.00 |
| Horiz. Flips | 0.99 | 1.55 | 1.79 | 1.84 | 1.85 | 1.79 | 1.88 | - |
| Det. Horiz. Flips | 0.96 | 1.89 | - | - | - | - | - | - |
| Vert. Flips | 0.68 | 0.94 | 1.09 | 1.23 | 1.17 | 1.14 | 1.25 | 1.20 |
| Det. Vert. Flips | 0.05 | 1.30 | - | - | - | - | - | - |
| Random Crops | 0.98 | 1.90 | 2.36 | 2.54 | 2.61 | 2.74 | 2.59 | 2.74 |
| Flips&Crops | 0.99 | 1.93 | 2.94 | 3.72 | 4.00* | 4.00* | 3.79 | - |
| Perspectives | 0.88 | 1.53 | 1.89 | 2.29 | 2.54 | 2.68 | 2.50 | - |
| Jitter | 0.91 | 0.92 | 0.90 | 0.87 | 0.82 | 0.81 | 0.93 | 0.88 |
| Blur | 0.76 | 0.76 | 0.75 | 0.70 | 0.66 | 0.62 | 0.76 | 0.69 |
| AutoAug | 0.78 | 0.95 | 1.02 | 1.20 | 1.39 | 1.52 | 1.63 | 1.77 |
| AugMix | 0.87 | 0.95 | 0.98 | 1.00 | 1.02 | 1.00 | 1.14 | 1.13 |
| RandAug | 0.88 | 1.49 | 1.91 | 2.20 | 2.51 | 2.67 | 2.49 | - |
| TrivialAug | 0.72 | 0.96 | 1.23 | 1.50 | 1.70 | 1.87 | 2.12 | - |
| AutoAug&Flips&Crops | 0.75 | 1.43 | 2.22 | 3.21 | 4.00* | 4.00* | 4.00* | 4.08* |
| Augmix&Flips&Crops | 0.86 | 1.62 | 2.50 | 3.09 | 3.82 | 3.84 | 3.74 | 3.74 |
| RandAug&Flips&Crops | 0.84 | 1.71 | 2.65 | 3.78 | 4.00* | 4.00* | 4.00* | - |
| TrivialAug&Flips&Crops | 0.70 | 1.31 | 1.98 | 2.86 | 3.71 | 4.00* | 4.00* | - |

**Table 2:** Extended table of **Exchange rates** for augmentations applied to 48000 base samples from the CINIC-10 training set, compared to reference models trained without augmentations on up to 192000 samples. We measure the exchange rate w.r.t. accuracy on the **CIFAR-10 val. set**. Values marked with ∗ fall outside the range of reference datasets and are extrapolated using power laws.

| CIFAR-10 (slightly out-of-domain) | | | | | | | |
|---|---|---|---|---|---|---|---|
| Augmentation | 1x | 2x | 4x | 8x | 16x | 32x | rand (160) |
| - | 1.00 | 1.00 | 1.00 | 1.00 | 1.00 | 1.00 | 1.00 |
| Horiz. Flips | 0.95 | 1.34 | 1.58 | 1.37 | 1.42 | 1.35 | 1.66 |
| Det. Horiz. Flips | 0.95 | 1.46 | - | - | - | - | - |
| Vert. Flips | 0.47 | 0.56 | 0.62 | 0.64 | 0.64 | 0.66 | 0.68 |
| Det. Vert. Flips | 0.02* | 0.71 | - | - | - | - | - |
| Random Crops | 0.94 | 1.82 | 1.93 | 1.91 | 1.75 | 1.92 | 1.91 |
| Flips&Crops | 0.96 | 1.78 | 2.15 | 2.58 | 2.26 | 3.05 | 1.94 |
| Perspectives | 0.95 | 2.06 | 3.29 | 4.02* | 4.73* | 4.96* | 4.34* |
| Jitter | 0.97 | 1.04 | 1.09 | 0.95 | 0.89 | 0.86 | 1.08 |
| Blur | 1.52 | 1.44 | 1.35 | 1.20 | 1.03 | 0.97 | 1.40 |
| AutoAug | 0.99 | 1.60 | 2.00 | 2.39 | 3.14 | 3.46 | 4.00* |
| AugMix | 1.77 | 2.38 | 2.61 | 3.10 | 3.18 | 3.20 | 3.31 |
| RandAug | 1.15 | 2.29 | 3.42 | 4.00* | 4.56* | 5.19* | 4.02* |
| TrivialAug | 0.89 | 1.81 | 2.30 | 3.17 | 4.00* | 4.02* | 4.78* |
| AutoAug&Flips&Crops | 0.96 | 2.53 | 4.46* | 6.30* | 6.93* | 7.27* | 7.18* |
| AugMix&Flips&Crops | 1.74 | 4.41* | 6.86* | 8.66* | 8.92* | 9.45* | 9.01* |
| RandAug&Flips&Crops | 0.96 | 2.32 | 4.00* | 5.10* | 6.24* | 6.80* | 5.12* |
| TrivialAug&Flips&Crops | 0.93 | 2.43 | 4.30* | 6.10* | 6.84* | 7.34* | 7.60* |

**Table 3:** Extended table of **Exchange rates** for augmentations applied to 48000 base samples from the CINIC-10 training set, compared to reference models trained without augmentations on up to 192000 samples. We measure the exchange rate w.r.t. accuracy on the **CIFAR-10-C val. set**. Values marked with ∗ fall outside the range of reference datasets and are extrapolated using power laws. Note that especially values $> 10$ are an extensive extrapolation far outside the measured range. Values marked with ✓ are outside the range of the estimated power law, meaning that (at least according to the behavior predicted by it), no amount of additional real data with be sufficient to match the accuracy achieved with this augmentation - there is no exchange rate.

| CIFAR-10-C (out-of-domain) | | | | | | | |
|---|---|---|---|---|---|---|---|
| Augmentation | 1x | 2x | 4x | 8x | 16x | 32x | rand (160) |
| - | 1.00 | 1.00 | 1.00 | 1.00 | 1.00 | 1.00 | 1.00 |
| Horiz. Flips | 0.84 | 0.67 | 0.66 | 0.55 | 0.53 | 0.52 | 0.63 |
| Det. Horiz. Flips | 0.93 | 0.55 | - | - | - | - | - |
| Vert. Flips | 0.15 | 0.11 | 0.11 | 0.11 | 0.11 | 0.12 | 0.11 |
| Det. Vert. Flips | 0.02* | 0.12 | - | - | - | - | - |
| Random Crops | 0.82 | 0.86 | 0.70 | 0.66 | 0.60 | 0.69 | 0.67 |
| Flips&Crops | 0.86 | 0.66 | 0.63 | 0.78 | 0.64 | 0.71 | 0.25 |
| Perspectives | 34.16* | ✓ | ✓ | ✓ | ✓ | ✓ | ✓ |
| Jitter | 16.81* | 190.76* | ✓ | 16.99* | 7.26* | 3.08 | 163.08* |
| Blur | ✓ | ✓ | ✓ | ✓ | ✓ | ✓ | ✓ |
| AutoAug | ✓ | ✓ | ✓ | ✓ | ✓ | ✓ | ✓ |
| AugMix | ✓ | ✓ | ✓ | ✓ | ✓ | ✓ | ✓ |
| RandAug | ✓ | ✓ | ✓ | ✓ | ✓ | ✓ | ✓ |
| TrivialAug | ✓ | ✓ | ✓ | ✓ | ✓ | ✓ | ✓ |
| AutoAug&Flips&Crops | ✓ | ✓ | ✓ | ✓ | ✓ | ✓ | ✓ |
| AugMix&Flips&Crops | ✓ | ✓ | ✓ | ✓ | ✓ | ✓ | ✓ |
| RandAug&Flips&Crops | 91.64* | ✓ | ✓ | ✓ | ✓ | ✓ | ✓ |
| TrivialAug&Flips&Crops | ✓ | ✓ | ✓ | ✓ | ✓ | ✓ | ✓ |

**Table 4: End-of-training stochasticity correlates strongly with flatness.** Gradient standard deviation across batches at the end of training and flatness measurements for ResNet-18 models trained on CIFAR-10 with various augmentations and strategies for sampling augmented views.

| Augmentation | Fixed Views | Same Batch | Grad. Std. | Flatness |
|---|---|---|---|---|
| No Augmentation | - | - | 13.201 | 11.4192 |
| Flips & Crops | ✗ | ✗ | 24.7397 | 16.3284 |
| | ✓ | ✗ | 13.612 | 10.1113 |
| | ✗ | ✓ | 15.676 | 11.914 |
| | ✓ | ✓ | 10.263 | 8.2033 |
| TrivialAug &Flips & Crops | ✗ | ✗ | 31.339 | 20.1336 |
| | ✓ | ✗ | 22.689 | 16.3189 |
| | ✗ | ✓ | 25.53 | 16.1076 |
| | ✓ | ✓ | 14.105 | 10.4287 |
| RandAug & Horiz. Flip & Random Crop | ✗ | ✗ | 29.912 | 18.6886 |
| | ✓ | ✗ | 16.585 | 11.5707 |
| | ✗ | ✓ | 22.0127 | 13.3998 |
| | ✓ | ✓ | 10.865 | 8.1097 |

**Table 5:** Extended table of **Exchange rates** for augmentations applied to 48000 base samples from the CINIC-10 training set, compared to reference models trained without augmentations on up to 192000 samples for **ResNet-8** models. We measure the exchange rate w.r.t. accuracy on the **CINIC-10 val. set**. Values marked with ∗ fall outside the range of reference datasets and are extrapolated using power laws.

| Augmentation | CINIC-10 (in-domain) | | | | | | | |
| | 1x | 2x | 4x | 8x | 16x | 32x | rand (160) | rand (640) |
|---|---|---|---|---|---|---|---|---|
| - | 1.00 | 1.00 | 1.00 | 1.00 | 1.00 | 1.00 | 1.00 | 1.00 |
| Horiz. Flips | 1.02 | 1.50 | 1.74 | 1.78 | 1.69 | 1.67 | 1.91 | 1.69 |
| Det. Horiz. Flips | 1.05 | 2.03 | - | - | - | - | - | - |
| Vert. Flips | 0.68 | 0.92 | 1.10 | 1.03 | 1.02 | 1.00 | 1.22 | 1.09 |
| Det. Vert. Flips | 0.09 | 1.31 | - | - | - | - | - | - |
| Random Crops | 0.98 | 1.82 | 2.40 | 2.44 | 2.51 | 2.62 | - | 2.62 |
| Flips&Crops | 0.96 | 1.89 | 2.69 | 2.99 | 3.33 | 4.00* | 1.07 | 3.83 |
| Perspectives | 0.90 | 1.57 | 1.95 | 2.13 | 2.29 | 2.18 | 2.39 | 2.35 |
| Jitter | 1.00 | 1.15 | 1.18 | 1.05 | 1.06 | 1.04 | 1.24 | 1.21 |
| Blur | 0.77 | 0.90 | 0.98 | 0.96 | 0.96 | 0.93 | 1.05 | 0.97 |
| AutoAug | 0.93 | 1.32 | 1.64 | 1.64 | 1.69 | 1.68 | 1.91 | 1.80 |
| AugMix | 1.03 | 1.31 | 1.52 | 1.54 | 1.54 | 1.48 | 1.75 | 1.67 |
| RandAug | 0.97 | 1.66 | 2.06 | 2.26 | 2.42 | 2.45 | 2.67 | 2.68 |
| TrivialAug | 0.82 | 1.39 | 1.84 | 1.96 | 2.07 | 1.99 | 2.19 | 2.24 |
| AutoAug&Flips&Crops | 0.85 | 1.62 | 2.15 | 2.65 | 2.89 | 3.14 | 2.62 | 3.00 |
| AugMix&Flips&Crops | 0.92 | 1.75 | 2.44 | 2.79 | 3.08 | 3.45 | 2.82 | 3.22 |
| RandAug&Flips&Crops | 0.93 | 1.78 | 2.47 | 2.84 | 3.28 | 4.00* | 2.84 | 3.92 |
| TrivialAug&Flips&Crops | 0.75 | 1.62 | 2.03 | 2.51 | 2.75 | 2.91 | 2.52 | 2.93 |

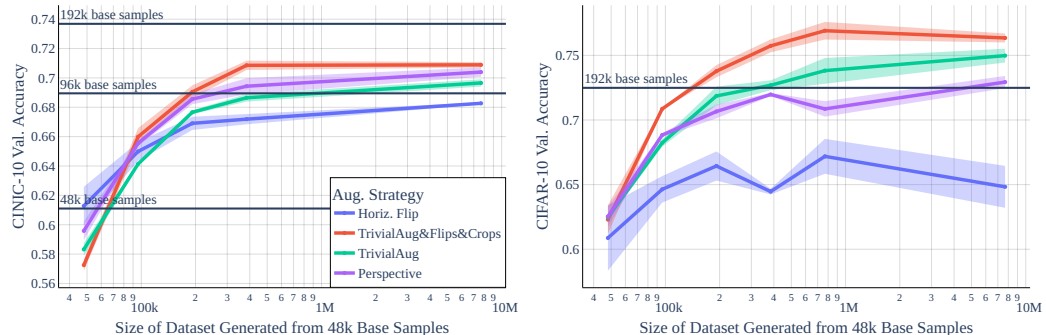

**Figure 14:** Validation accuracy versus dataset size as **larger datasets are generated from a fixed number of base samples** and selected data augmentations. **ResNet-8** models are trained on fixed datasets generated via augmentation from 48000 base samples from the CINIC-10 train set and evaluated on the CINIC-10 val. set (**left**) and the CIFAR-10 val. set (**right**), std. error over 3 runs shaded. The accuracy of reference models trained without augmentations is marked with horizontal lines.

**Table 6:** Extended table of **Exchange rates** for augmentations applied to 48000 base samples from the CINIC-10 training set, compared to reference models trained without augmentations on up to 192000 samples for **VGG-11** models. We measure the exchange rate w.r.t. accuracy on the **CINIC-10 val. set**. Values marked with ∗ fall outside the range of reference datasets and are extrapolated using power laws.

| Augmentation | CINIC-10 (in-domain) | | | | | | | |
| --- | --- | --- | --- | --- | --- | --- | --- | --- |
| | 1x | 2x | 4x | 8x | 16x | 32x | rand (160) | rand (640) |
| - | 1.00 | 1.00 | 1.00 | 1.00 | 1.00 | 1.00 | 1.00 | 1.00 |
| Horiz. Flips | 1.00 | 1.54 | 1.77 | 1.78 | 1.77 | 1.78 | 1.89 | 1.86 |
| Det. Horiz. Flips | 0.98 | 1.97 | - | - | - | - | - | - |
| Vert. Flips | 0.54 | 0.83 | 0.93 | 0.95 | 0.93 | 0.92 | 1.00 | 0.97 |
| Det. Vert. Flips | 0.02 | 1.08 | - | - | - | - | - | - |
| Random Crops | 0.98 | 1.70 | 2.11 | 2.12 | 2.22 | 2.11 | - | 2.24 |
| Flips&Crops | 0.98 | 1.86 | 2.56 | 2.99 | 3.08 | 3.11 | 1.01 | 3.15 |
| Perspectives | 0.79 | 1.20 | 1.54 | 1.56 | 1.51 | 1.46 | 1.76 | 1.51 |
| Jitter | 0.89 | 0.86 | 0.86 | 0.84 | 0.79 | 0.81 | 0.86 | 0.82 |
| Blur | 0.60 | 0.59 | 0.58 | 0.54 | 0.56 | 0.51 | 0.62 | 0.55 |
| AutoAug | 0.72 | 0.91 | 1.05 | 1.10 | 1.26 | 1.24 | 1.59 | 1.51 |
| AugMix | 0.91 | 0.97 | 0.97 | 0.96 | 0.95 | 0.97 | 1.02 | 1.00 |
| RandAug | 0.83 | 1.37 | 1.79 | 1.92 | 2.00 | 1.98 | 2.14 | 2.19 |
| TrivialAug | 0.64 | 0.94 | 1.14 | 1.37 | 1.55 | 1.65 | 1.94 | 2.00 |
| AutoAug&Flips&Crops | 0.67 | 1.28 | 2.09 | 2.68 | 3.13 | 3.21 | 4.00* | 4.00* |
| AugMix&Flips&Crops | 0.83 | 1.55 | 2.31 | 2.79 | 3.06 | 2.81 | 3.22 | 3.12 |
| RandAug&Flips&Crops | 0.79 | 1.57 | 2.27 | 3.09 | 3.32 | 3.33 | 3.97 | 3.92 |
| TrivialAug&Flips&Crops | 0.54 | 1.13 | 1.83 | 2.46 | 2.93 | 3.35 | 4.00* | 4.00* |

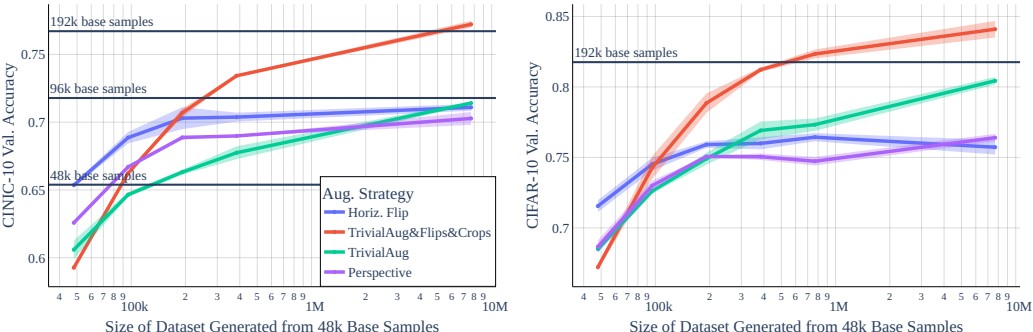

**Figure 15:** Validation accuracy versus dataset size as **larger datasets are generated from a fixed number of base samples** and selected data augmentations. **VGG-11** models are trained on fixed datasets generated via augmentation from 48000 base samples from the CINIC-10 train set and evaluated on the CINIC-10 val. set (**left**) and the CIFAR-10 val. set (**right**), std. error over 3 runs shaded. The accuracy of reference models trained without augmentations is marked with horizontal lines.

**Table 7:** Extended table of **Exchange rates** for augmentations applied to 48000 base samples from the CINIC-10 training set, compared to reference models trained without augmentations on up to 192000 samples for **ConvMixer** models. We measure the exchange rate w.r.t. accuracy on the **CINIC-10 val. set**. Values marked with ∗ fall outside the range of reference datasets and are extrapolated using power laws.

| | **CINIC-10** (in-domain) | | | | | | | |
| Augmentation | 1x | 2x | 4x | 8x | 16x | 32x | rand (160) | rand (640) |
|---|---|---|---|---|---|---|---|---|
| - | 1.00 | 1.00 | 1.00 | 1.00 | 1.00 | 1.00 | 1.00 | 1.00 |
| Horiz. Flips | 0.95 | 1.37 | 1.39 | 1.59 | 1.50 | 1.69 | 1.41 | 1.66 |
| Det. Horiz. Flips | 1.04 | 1.50 | - | - | - | - | - | - |
| Vert. Flips | 0.37 | 0.45 | 0.45 | 0.48 | 0.55 | 0.50 | 0.48 | 0.49 |
| Det. Vert. Flips | 0.04 | 0.43 | - | - | - | - | - | - |
| Random Crops | 0.78 | 0.93 | 0.97 | 1.18 | 1.36 | 1.30 | 1.00 | 1.26 |
| Flips&Crops | 0.87 | 1.05 | 1.40 | 1.71 | 1.59 | 1.63 | 1.00 | 1.72 |
| Perspectives | 0.71 | 0.97 | 1.26 | 1.85 | 2.24 | 2.03 | 2.04 | 2.33 |
| Jitter | 0.78 | 0.91 | 0.90 | 1.04 | 1.11 | 1.34 | 0.87 | 1.26 |
| Blur | 0.34 | 0.46 | 0.51 | 0.49 | 0.56 | 0.61 | 0.54 | 0.65 |
| AutoAug | 0.59 | 0.89 | 1.42 | 1.77 | 1.96 | 2.30 | 1.57 | 2.07 |
| AugMix | 0.77 | 0.91 | 0.89 | 1.31 | 1.60 | 1.77 | 1.40 | 1.70 |
| RandAug | 0.80 | 1.28 | 1.71 | 1.76 | 2.05 | 2.59 | 2.21 | 2.28 |
| TrivialAug | 0.48 | 1.11 | 1.72 | 2.31 | 2.84 | 2.91 | 2.28 | 2.66 |
| AutoAug&Flips&Crops | 0.49 | 1.04 | 1.85 | 2.19 | 2.67 | 2.80 | 2.49 | 2.91 |
| AugMix&Flips&Crops | 0.65 | 1.22 | 1.58 | 1.91 | 2.10 | 2.12 | 1.92 | 2.20 |
| RandAug&Flips&Crops | 0.57 | 1.19 | 1.82 | 2.40 | 2.65 | 2.69 | 2.89 | 2.89 |
| TrivialAug&Flips&Crops | 0.41 | 1.07 | 1.83 | 2.21 | 2.92 | 3.55 | 2.67 | 3.08 |

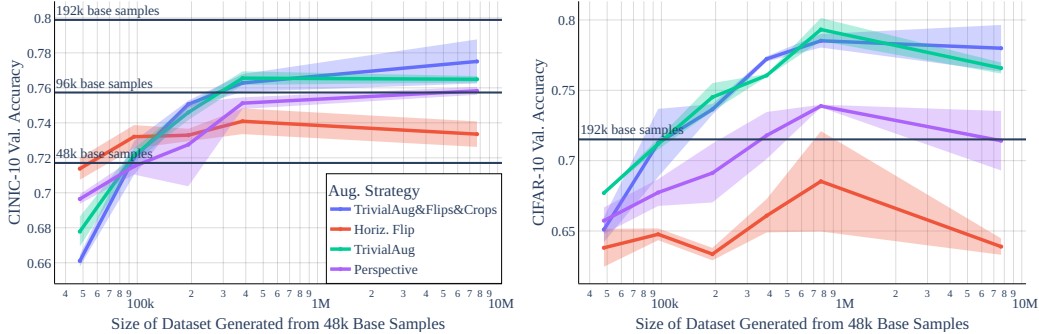

**Figure 16:** Validation accuracy versus dataset size as **larger datasets are generated from a fixed number of base samples** and selected data augmentations. **ConvMixer** models are trained on fixed datasets generated via augmentation from 48000 base samples from the CINIC-10 train set and evaluated on the CINIC-10 val. set (**left**) and the CIFAR-10 val. set (**right**), std. error over 3 runs shaded. The accuracy of reference models trained without augmentations is marked with horizontal lines.

**Table 8:** Extended table of **Exchange rates** for augmentations applied to 48000 base samples from the **MNIST** training set, compared to reference models trained without augmentations on up to 60000 samples for **ResNet-18** models. We measure the exchange rate w.r.t. accuracy on the **MNIST val. set**. Values marked with ∗ fall outside the range of reference datasets and are extrapolated using power laws. Values marked with ✓ are outside the range of the estimated power law, meaning that (at least according to the behavior predicted by it), no amount of additional real data with be sufficient to match the accuracy achieved with this augmentation - there is no exchange rate.

| Augmentation | MNIST (in-domain) | | | | | | | |
| --- | --- | --- | --- | --- | --- | --- | --- | --- |
| | 1x | 2x | 4x | 8x | 16x | 32x | rand (160) | rand (640) |
| - | 1.00 | 1.00 | 1.00 | 1.00 | 1.00 | 1.00 | 1.00 | 1.00 |
| Horiz. Flips | 0.24 | 0.24 | 0.34 | 0.27 | 0.26 | 0.29 | 0.40 | 0.25 |
| Det. Horiz. Flips | 0.02* | - | - | - | - | - | - | - |
| Vert. Flips | 0.20 | 0.22 | 0.22 | 0.22 | 0.22 | 0.24 | 0.23 | 0.22 |
| Det. Vert. Flips | 0.02* | - | - | - | - | - | - | - |
| Random Crops | 1.11 | 1.16 | ✓ | 1.17 | ✓ | ✓ | - | 37.32* |
| Flips&Crops | 0.17 | 0.23 | 0.21 | 0.22 | 0.24 | 0.25 | 0.88 | 0.23 |
| Perspectives | 1.16 | 1.14 | 7.84* | 1.17 | 1.17 | ✓ | 1.15 | 37.32* |
| Jitter | 1.08 | 0.81 | 1.17 | 1.08 | 1.11 | 1.15 | 0.81 | 1.10 |
| Blur | 1.08 | 1.10 | 1.12 | 0.77 | 1.15 | 7.84* | 1.17 | 1.14 |
| AutoAug | 0.72 | 1.16 | 1.08 | 1.16 | 4.38* | 7.84* | 7.84* | ✓ |
| AugMix | 1.11 | 1.14 | 1.16 | 1.16 | 1.17 | 1.17 | 1.12 | 1.15 |
| RandAug | 1.10 | 1.15 | 4.38* | ✓ | ✓ | ✓ | 1.16 | ✓ |
| TrivialAug | 0.54 | 4.38* | 1.12 | 1.12 | 4.38* | ✓ | 1.16 | ✓ |
| AutoAug&Flips&Crops | 0.14 | 0.17 | 0.20 | 0.24 | 0.26 | 0.24 | 0.24 | 0.26 |
| AugMix&Flips&Crops | 0.15 | 0.21 | 0.21 | 0.24 | 0.23 | 0.24 | 0.24 | 0.24 |
| RandAug&Flips&Crops | 0.17 | 0.21 | 0.21 | 0.23 | 0.24 | - | 0.23 | 0.23 |
| TrivialAug&Flips&Crops | 0.12 | 0.15 | 0.16 | 0.21 | 0.22 | 0.23 | 0.21 | 0.24 |

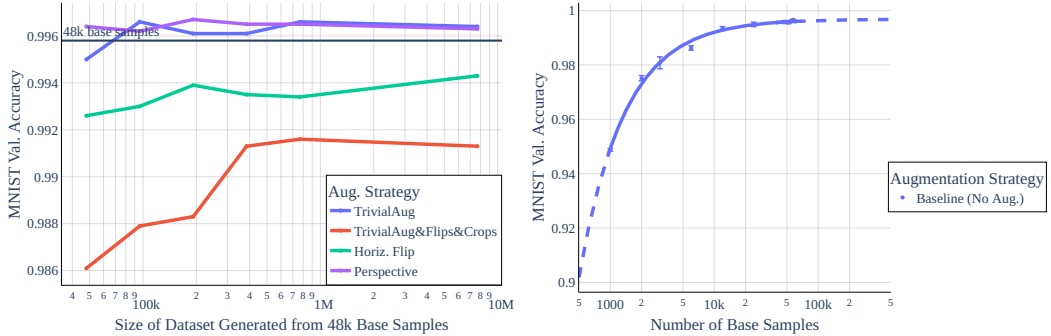

**Figure 17: Left:** Validation accuracy versus dataset size as **larger datasets are generated from a fixed number of base samples** and selected data augmentations. **ResNet-18** models are trained on fixed datasets generated via augmentation from 48000 base samples from the MNIST train set and evaluated on the MNIST val. set. **Right:** Extrapolated scaling behavior of reference models for MNIST.

**Table 9:** Extended table of **Exchange rates** for augmentations applied to 48000 base samples from the **CIFAR-100** training set, compared to reference models trained without augmentations on up to 50000 samples for **ResNet-18** models. We measure the exchange rate w.r.t. accuracy on the **CIFAR-100 val. set**. Values marked with ∗ fall outside the range of reference datasets and are extrapolated using power laws.

| Augmentation | **CIFAR-100** (in-domain) | | | | | | | |
| | 1x | 2x | 4x | 8x | 16x | 32x | rand (160) | rand (640) |
| --- | --- | --- | --- | --- | --- | --- | --- | --- |
| - | 1.00 | 1.00 | 1.00 | 1.00 | 1.00 | 1.00 | 1.00 | 1.00 |
| Horiz. Flips | 0.95 | 1.39* | 1.57* | 1.59* | 1.56* | 1.52* | 1.61* | 1.55* |
| Det. Horiz. Flips | 0.89 | 1.64* | - | - | - | - | - | - |
| Vert. Flips | 0.62 | 0.94 | 1.13* | 1.14* | 1.11* | 1.02 | 1.18* | 1.14* |
| Det. Vert. Flips | 0.18 | 1.19* | - | - | - | - | - | - |
| Random Crops | 0.90 | 1.58* | 1.88* | 2.00* | 1.99* | 1.99* | - | 2.04* |
| Flips&Crops | 0.87 | 1.60* | 2.08* | 2.30* | 2.35* | 2.28* | 0.99 | 2.35* |
| Perspectives | 0.78 | 1.33* | 1.66* | 1.90* | 1.97* | 1.94* | 1.87* | 1.96* |
| Jitter | 0.88 | 0.90 | 0.94 | 0.88 | 0.82 | 0.80 | 0.90 | 0.82 |
| Blur | 0.77 | 0.75 | 0.74 | 0.70 | 0.67 | 0.67 | 0.73 | 0.69 |
| AutoAug | 0.71 | 0.92 | 0.99 | 1.01 | 1.10* | 1.14* | 1.40* | 1.37* |
| AugMix | 0.86 | 0.97 | 1.00 | 1.02 | 1.03 | 1.02 | 1.15* | 1.14* |
| RandAug | 0.79 | 1.30* | 1.58* | 1.80* | 1.86* | 1.88* | 1.81* | 1.87* |
| TrivialAug | 0.59 | 0.91 | 1.12* | 1.21* | 1.32* | 1.48* | 1.78* | 1.86* |
| AutoAug&Flips&Crops | 0.60 | 1.22* | 1.73* | 2.04* | 2.20* | 2.23* | 2.29* | 2.37* |
| AugMix&Flips&Crops | 0.75 | 1.47* | 1.94* | 2.23* | 2.26* | 2.30* | 2.15* | 2.20* |
| RandAug&Flips&Crops | 0.67 | 1.39* | 1.94* | 2.23* | 2.26* | 2.31* | 2.24* | 2.24* |
| TrivialAug&Flips&Crops | 0.51 | 1.03 | 1.58* | 1.92* | 2.10* | 2.22* | 2.37* | 2.55* |

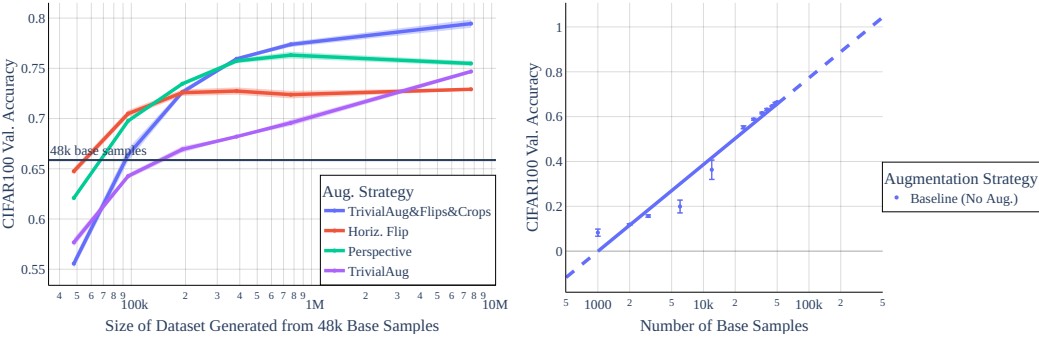

**Figure 18: Left:** Validation accuracy versus dataset size as **larger datasets are generated from a fixed number of base samples** and selected data augmentations. **ResNet-18** models are trained on fixed datasets generated via augmentation from 48000 base samples from the CIFAR-100 train set and evaluated on the CIFAR-100 val. set. **Right:** Extrapolated scaling behavior of reference models for CIFAR-100.

**Table 10:** Extended table of **Exchange rates** for augmentations applied to 48000 base samples from the **EMNIST** training set, compared to reference models trained without augmentations on up to 124800 samples for **ResNet-18** models. We measure the exchange rate w.r.t. accuracy on the **EMNIST val. set**. Values marked with ∗ fall outside the range of reference datasets and are extrapolated using power laws. Values marked with ✓ are outside the range of the estimated power law, meaning that (at least according to the behavior predicted by it), no amount of additional real data with be sufficient to match the accuracy achieved with this augmentation - there is no exchange rate.

| Augmentation | **EMNIST** (in-domain) | | | | | | | |
|---|---|---|---|---|---|---|---|---|
| | 1x | 2x | 4x | 8x | 16x | 32x | rand (160) | rand (640) |
| - | 1.00 | 1.00 | 1.00 | 1.00 | 1.00 | 1.00 | 1.00 | 1.00 |
| Horiz. Flips | 0.12 | 0.15 | 0.15 | 0.16 | 0.15 | 0.15 | 0.16 | 0.16 |
| Det. Horiz. Flips | 0.02* | - | - | - | - | - | - | - |
| Vert. Flips | 0.12 | 0.16 | 0.17 | 0.18 | 0.16 | 0.17 | 0.17 | 0.17 |
| Det. Vert. Flips | 0.02* | - | - | - | - | - | - | - |
| Random Crops | 0.61 | 0.93 | 1.90 | 2.10 | 5.90* | ✓ | - | ✓ |
| Flips&Crops | 0.10 | 0.12 | 0.15 | 0.18 | 0.19 | 0.23 | 1.12 | 0.20 |
| Perspectives | 0.95 | 2.04 | 1.99 | 2.11 | 2.06 | ✓ | ✓ | 2.18 |
| Jitter | 0.72 | 1.45 | 0.90 | 0.91 | 1.36 | 0.98 | 1.03 | 1.24 |
| Blur | 0.82 | 0.81 | 0.75 | 0.88 | 0.94 | 0.93 | 0.92 | 1.16 |
| AutoAug | 1.24 | 1.26 | 0.99 | 0.92 | 2.12 | 2.07 | 13.79* | 2.03 |
| AugMix | 1.60 | 0.90 | 1.04 | 2.10 | 2.04 | 2.02 | 1.56 | 1.99 |
| RandAug | 1.24 | 2.14 | 2.05 | 2.07 | ✓ | ✓ | ✓ | ✓ |
| TrivialAug | 0.86 | 0.79 | 0.96 | 1.91 | 1.57 | 1.84 | ✓ | ✓ |
| AutoAug&Flips&Crops | 0.10 | 0.12 | 0.12 | 0.14 | 0.21 | 0.21 | 0.30 | 0.25 |
| AugMix&Flips&Crops | 0.10 | 0.12 | 0.12 | 0.17 | 0.17 | 0.19 | 0.21 | 0.16 |
| RandAug&Flips&Crops | 0.10 | 0.13 | 0.15 | 0.18 | 0.20 | 0.20 | 0.24 | 0.21 |
| TrivialAug&Flips&Crops | 0.06 | 0.10 | 0.11 | 0.15 | 0.20 | 0.22 | 0.28 | 0.23 |

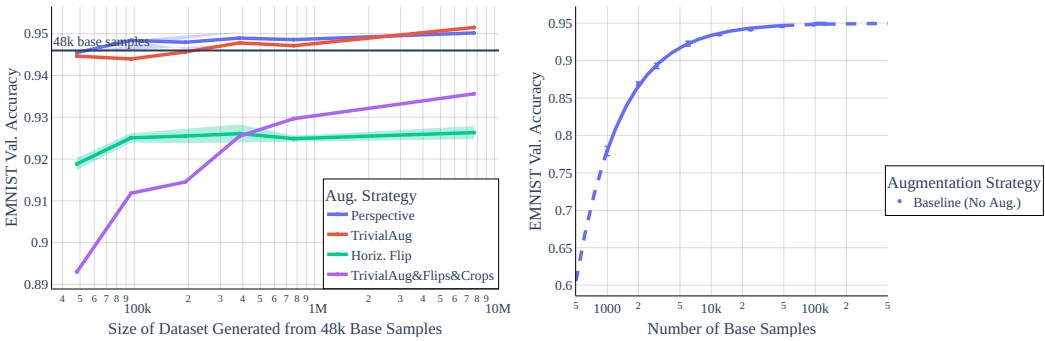

**Figure 19: Left:** Validation accuracy versus dataset size as **larger datasets are generated from a fixed number of base samples** and selected data augmentations. **ResNet-18** models are trained on fixed datasets generated via augmentation from 48000 base samples from the EMNIST train set and evaluated on the EMNIST val. set. **Right:** Extrapolated scaling behavior of reference models for EMNIST.

**Table 11:** Gradient standard deviation across batches at the end of training and flatness measurements for Mobilenet V2 models trained on CIFAR-100 with various augmentations and strategies for sampling augmented views. Averaged over 3 runs

| Augmentation | Fixed Views | Same Batch | Grad. Std. | Flatness |
|---|---|---|---|---|
| No Augmentation | - | - | 46.39 | 7.78 |
| Horiz. Flip & Rand. Crop | No | No | 46.37 | 7.79 |
| | Yes | No | 42.62 | 3.41 |

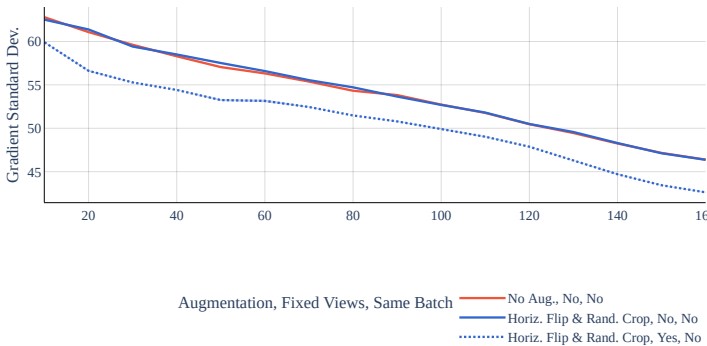

**Figure 20:** Standard deviation of gradient across epochs for different augmentations and different mini-batch sampling strategies. Each dot indicates the mean over 3 runs, and shaded regions represent confidence intervals of width one standard error.

## D   ALTERNATIVE POWER LAWS

In the main body, we propose to investigate power laws of the form $f(x) = ax^{-c} + b$ and fit these to measured data to estimate scaling rates for accuracy with respect to number of samples. We believe this choice to be sound in principle, as over all variants, e.g. in Figure 1 (left), these curves fit experimental results well. In this section, though, we want to validate this choice and discuss possible alternatives.

### D.1   EQUATIONS SUGGESTED VIA SYMBOLIC REGRESSION

To potentially discover alternative functional forms we turn to symbolic regression using machine learning scoring rules (Cranmer et al., 2020) using the implementation of Cranmer (2022). We search for symbolic expressions containing elementary operations and exponential functions with a symbolic complexity less or equal than 10. For exponentiation, we limit complexity in the exponent to 1. We search for 24000 iterations and then extract the functional form of the equation $f$, discarding the constants obtained during discovery, and use standard nonlinear regression, as described previously to fit constants for each curve $f$ and $f_{\text{aug}}$. We symbolically invert the found equation $f$ and compute $f^{-1}(f_{\text{aug}})(x) - x$ as described in Section 3.

To run a representative example with higher complexity, we re-evaluate Figure 1 and discover a new functional form as described above using data from baseline experiments without augmentation. We first note that we refind a result close to our original hypothesis with $f_{\text{sym}}(x) = x^{0.061279207} - 1.2693417$, at complexity 5. Yet, we find the following function with complexity 10 using symbolic regression:

$$f_{\text{sym}}(x) = 0.897637144733759 e^{-\frac{1.34957448025087}{(0.000168170796495114x+1)^{0.761082360343512}}},$$

i.e. the functional form

$$f(x) = a \exp^{-\frac{b}{(cx+1)^d}}.$$

We plot results for a re-analysis of Figure 1 with this form in Figure 21. Unsurprisingly, this is a better fit for the baseline without augmentations, data on which this function was found. Interestingly though, this does not lead to significant qualitative changes. The perspective transform is valued differently based on this new fit of the baseline, but trends are broadly consistent.

If we take this to the extreme, and search for a functional form specifically without atoms used previously (containing now only additions, subtractions, divisions and multiplications), we find

$$f_{\text{sym}}(x) = 0.866576773986552 - \frac{10798.9835603424}{x + 17236.4768553924},$$

from which we fit

$$f(x) = a - \frac{b}{x + c},$$

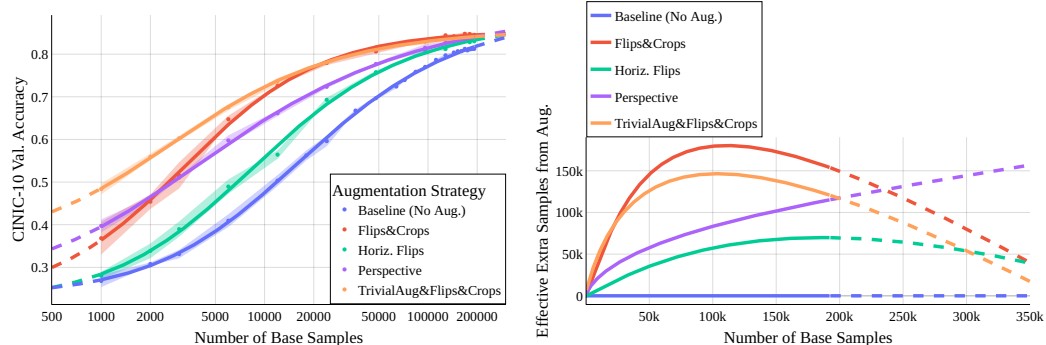

**Figure 21: Func. Form** $f(x) = a \exp^{-\frac{b}{(cx+1)^d}}$ found via symbolic regression for select augmentations applied randomly and the gain in terms of effective extra samples from Equation (1). Fitted curves marked in solid colors, with extrapolated regions dashed. **Left:** Number of base samples (from CINIC-10) on the logarithmic horizontal axis compared to validation accuracy. **Right**: Number of base samples compared to effective extra data, showing how the benefits of each data augmentation scale as the model is trained on more and more data.

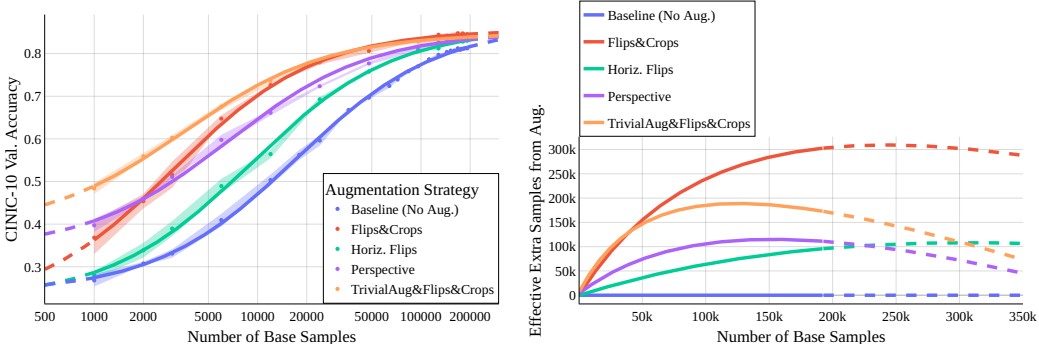

**Figure 22: Func. Form** $f(x) = a - \frac{b}{x+c}$ found via symbolic regression for select augmentations applied randomly and the gain in terms of effective extra samples from Equation (1). Fitted curves marked in solid colors, with extrapolated regions dashed. **Left:** Number of base samples (from CINIC-10) on the logarithmic horizontal axis compared to validation accuracy. **Right**: Number of base samples compared to effective extra data, showing how the benefits of each data augmentation scale as the model is trained on more and more data.

which we also use for an exemplary reanalysis and show in Figure 22. This agains fits the baseline very well, but for example fits the result with flips and crops less well than Figure 1.

We could ultimately also search for functions with higher complexity. For example, this is a symbolic regression result with larger complexity:

$$f_{\text{sym}} = 0.170768861592998^{\frac{0.617062576584116}{(0.000145101205003307x+1)^{0.768875806100543}}} - 0.103778477468959,$$

from which we extract

$$f_{\text{sym}} = a^{\frac{b}{(cx+1)^d}} - e$$

and which we visualize in Figure 23. Here, we note the close relationship to the previously found result at complexity 10, and similar conclusions hold.

## D.2 ALTERNATIVE SUGGESTIONS

Alternatively, we know that the functional form $f(x) = ax^{-c} + b$ can only locally describe the relationship between accuracy and data samples, as the resulting function is potentially unbounded above for some choices of $(a, b, c)$, yet accuracy is strictly bounded by $1.0$. As, such, we might wonder about fitting a globally accurate form, such as $f(x) = a \tanh(x^{-c}) + b$, which is bounded. We include a re-interpretation with this form in Figure 24, but again find no broad qualitative changes.

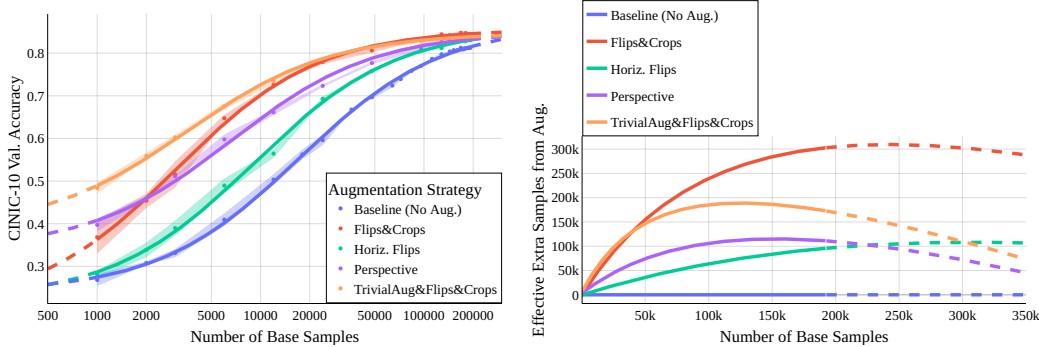

**Figure 23: Func. Form** $f(x) = a^{\frac{b}{(cx+1)^d}} - e$ found via symbolic regression for select augmentations applied randomly and the gain in terms of effective extra samples from Equation (1). Fitted curves marked in solid colors, with extrapolated regions dashed. **Left:** Number of base samples (from CINIC-10) on the logarithmic horizontal axis compared to validation accuracy. **Right**: Number of base samples compared to effective extra data, showing how the benefits of each data augmentation scale as the model is trained on more and more data.

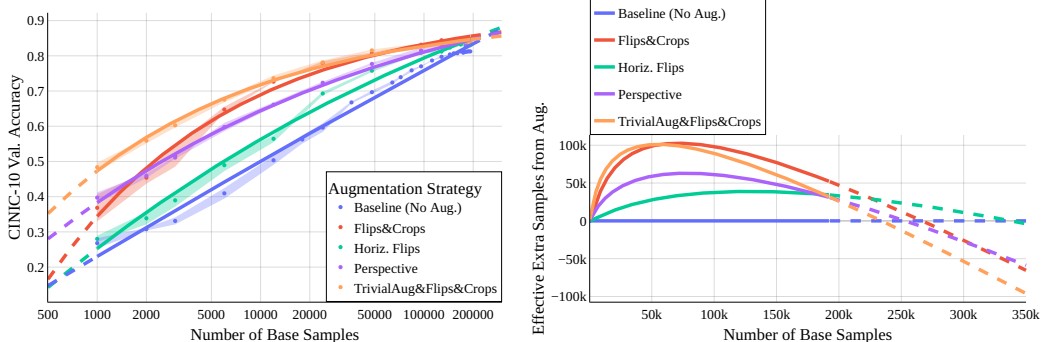

**Figure 24: Func. bounded Form** $f(x) = a \tanh\left(x^{-c}\right) + b$ found via symbolic regression for select augmentations applied randomly and the gain in terms of effective extra samples from Equation (1). Fitted curves marked in solid colors, with extrapolated regions dashed. **Left:** Number of base samples (from CINIC-10) on the logarithmic horizontal axis compared to validation accuracy. **Right**: Number of base samples compared to effective extra data, showing how the benefits of each data augmentation scale as the model is trained on more and more data.

## E  EXPERIMENTAL ABLATIONS

To validate the experimental setup executed in the main body of this work, we run a series of comparative studies. First, we construct a CINIC-10 setup in this work with a novel, sanitized test set that makes it hard to compare validation accuracies to baseline CIFAR-10 and CINIC-10. As such, we verify our implementations of all architectures and chosen budget and training setup in Table 12, where we find these implementations to perform as expected, and reach high performance on CIFAR-10. For reference, we also record the number of parameters for each investigated model here.

Secondly, we vary the budget used in our study. In all other experiments in this work we consider a budget of 60000 steps (which is 160 epochs at batch size 128 and a dataset size of 48000). In Figure 25, we check, for the case of TrivialAug, how the results of Figure 1 vary, as the budget is varied. There, we find that qualitative behavior is stable across all considered budget variations, but quantitative results depend on the chosen budget.

Third, we switch from validation accuracy after training to peak validation accuracy observed during training in Figure 26. Peak validation accuracy during training as an oracle would return early-stopping results, if the investigated models would overfit to their training data. However, we find in Figure 26 that behavior and analysis are almost indistinguishable under this metric, validating the choice of final validation accuracy without early stopping and the choice of fixed budgets for all dataset sizes.

| Model Architecture | Number of Parameters | CIFAR-10 Validation Accuracy |
|---|---|---|
| ResNet-18 (width 64) | 11 173 962 | 94.75 |
| ResNet-18 (width 4) | 44 622 | 78.25 |
| ResNet-18 (width 16) | 7 014 662 | 90.76 |
| ResNet-18 (width 128) | 44 662 922 | 95.08 |
| ResNet-18 (width 256) | 178 585 866 | 95.38 |
| ResNet-8 | 78 042 | 81.06 |
| ResNet-20 | 272 474 | 88.82 |
| ResNet-110 | 1 730 714 | 93.51 |
| ResNet-152 | 58 156 618 | 95.39 |
| PyramidNet-110 | 3 904 446 | 94.95 |
| VGG13 | 9 416 010 | 93.07 |
| SwinTransformer (v2) | 4 090 794 | 77.86 |
| ConvMixer | 1 277 962 | 94.57 |

**Table 12:** Baseline validation of our implementation. For each of the model architectures investigated in this work, we report number of parameters and validation accuracy of the model when trained on CIFAR-10 with the same experimental setup as described in the remainder of this work and augmentation policy TrivialAug & Random Crops & Flips.

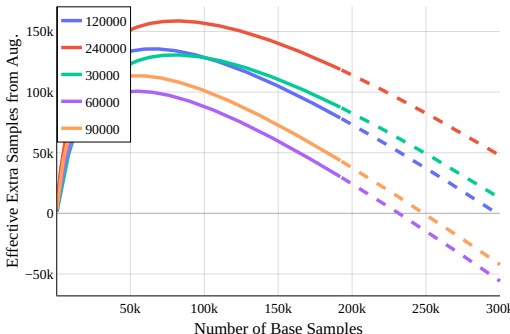

**Figure 25: Power laws**, Variant of Figure 1. Number of base samples compared to effective extra data, showing how the benefits of each TrivialAug scale as the model is trained on more data and vary as the model is trained with various step budgets.

We further include additional model architectures in Figure 27, validating that model family is a much stronger predictor of qualitative behavior than model size, by including additional results for the larger ResNet-110, ResNet-152, and the smaller ResNet-8 and ResNet-20.

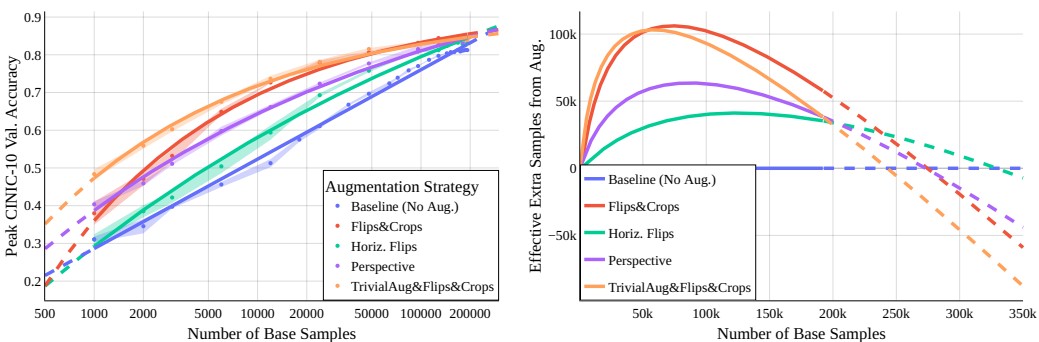

**Figure 26: Power laws based on Peak Val. Accuracy**, Variant of Figure 1. Evaluating $f(x) = ax^{-c} + b$ for select augmentations applied randomly and the gain in terms of effective extra samples from Equation (1). Fitted curves marked in solid colors, with extrapolated regions dashed. **Left:** Number of base samples (from CINIC-10) on the logarithmic horizontal axis compared to **peak validation accuracy**. **Right**: Number of base samples compared to effective extra data, showing how the benefits of each data augmentation scale as the model is trained on more and more data.

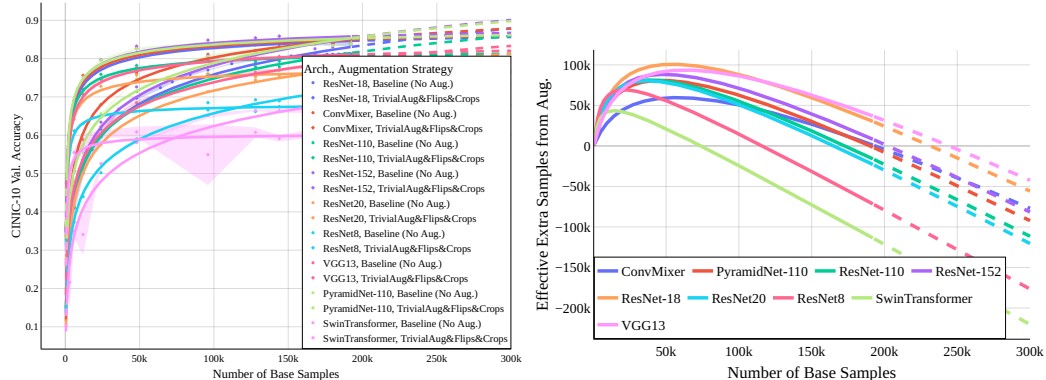

**Figure 27:** Extension of Figure 3. Power Laws and Exchange Rates via Equation (1) for *TrivialAug* when modifying more **model architectures**. **Left:** Accuracy power laws for various vision architectures. **Right:** Exchange Rates for select vision architectures. Exchange rates behave similarly for large classes of architectures.

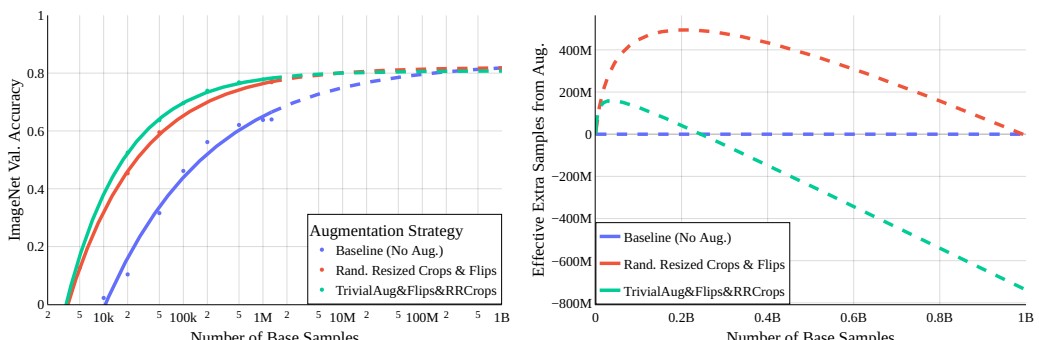

**Figure 28: Power laws for a ResNet-50 on ImageNet** $f(x) = ax^{-c} + b$ for select augmentations applied randomly and the gain in terms of effective extra samples from Equation (1). Fitted curves marked in solid colors, with extrapolated regions dashed. **Left:** Number of base samples (from ImageNet) on the logarithmic horizontal axis compared to validation accuracy. The scaling behavior of each augmentation is closely matched by these power laws. **Right**: Number of base samples compared to effective extra data, showing how the benefits of each data augmentation scale as the model is trained on more and more data.

# F IMAGENET RESULTS

We also investigate how well these power laws would fit ImageNet training runs (Russakovsky et al., 2015). ImageNet is not an ideal setting for our analysis, given that the dataset is relatively small compared to its complexity. Given unlimited compute one would rather train reference models on larger splits, e.g. ImageNet-21k or JFT-300m (Sun et al., 2017) and compare the validation accuracy of those reference models to ImageNet models trained with augmentations, as discussed in Section 3.

To train these models we follow the recently proposed training regime of Wightman et al. (2021). We train for 65536 steps (about 100 epochs) using the LAMB optimizer with a peak learning rate of $8e-3$ and cosine decay after a warmup of 3125 steps. We additionally apply label smoothing with 0.1. We train a ResNet-50 model as described in He et al. (2019a). As unaugmented baseline, we consider a pre-processing of centered crops of size 224. For the augmented variant, we augment with random resized crops (with ratios between 3/4 and 4/3 as usual) of size 224 and random horizontal flips. For TrivialAug, we use TrivialAug as described on top of these random resized crops and flips. In all cases, we validate by resizing to $256 \times 256$ and center cropping to $224 \times 224$.

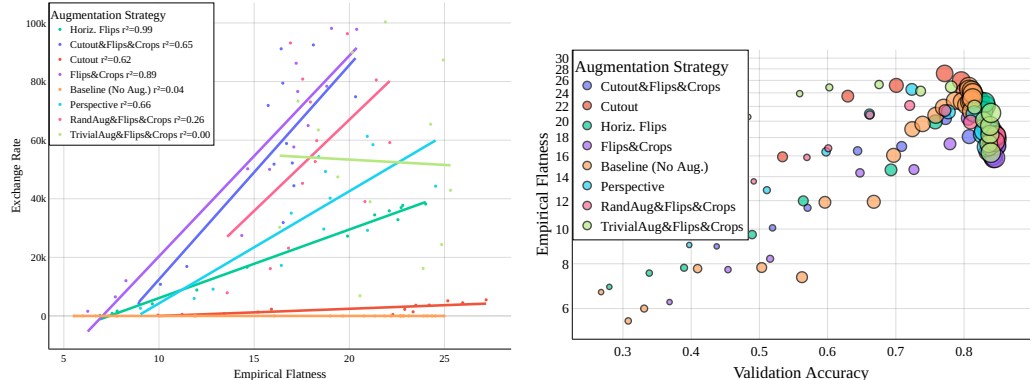

**Figure 29: Dataset scaling and flatness on CINIC-10** Direct measurements of flatness plotted measures of model performance on CINIC-10. Left: Flatness compared to effective exchange rates for a number of augmentation strategies and dataset sizes. Trend lines are ordinary linear regression, $r^2$ values are included in the legend. Right: Same data, but plotted against raw CINIC-10 validation accuracy. Dataset size is inverse-square proportional to marker size.

## G  REMARK: WELL-DEFINEDNESS OF EXCHANGE RATES

The exchange rate defined in Equation (1) relies on the functional form of both $f_{\text{ref}}$ and $f_{\text{aug}}$. To be more precise, in this work we would ideally assume that both functions are strictly monotonically increasing maps $f : [0, \infty) \to [0, 1)$ and surjective. However, both due to the simplicity of the chosen power laws (which are potentially unbounded), and actual observed data, both bounded range and surjectiveness can be violated in practice. As such, as $f^{-1}$ is only defined on $\text{Im}(f) \subset [0, 1)$, this can result in the exchange rate being only defined on a subset of sample sizes $S \subset [0, \infty)$. We can see this happening in Figure 4 (right), where we analyze an exchange on the out-of-domain dataset CIFAR-10-C (Hendrycks & Dietterich, 2018). Here, the exchange rate for e.g. TrivialAug with flips and crops is undefined for samples sizes greater than about 3000, with the exchange rate approaching infinity as samples sizes approach this limit.

While this may be inconvenient to visualize, we believe it to be an interesting feature of the concept of exchange rates. In the undefined range, there really is *no exchange* possible: Based on existing experimental data, even an extrapolation to infinite additional in-domain CINIC data is not enough to increase accuracy on out-of-domain CIFAR-10-C to the same value that is reached with the augmentation broadening the data distribution.

## H  DATASET SCALING AND FLATNESS

In extension of Section 5.3, we provide additional data in Figure 29. As discussed in the main body, we find a strong correlation in the number of samples gained through augmentation and flatness for all augmentation strategies except TrivialAug. We hone in on this trend as a function of dataset size on the right hand side of Figure 29. TrivialAug is possibly an outlier as it produces models that remain relatively flat for all sample sizes considered, as discussed in Section 5.3, and may lead to models that are "artificially" flat, but without correlation with accuracy or improved exchange rates.

