# OpenReview forum: "How Much Data Are Augmentations Worth?  An Investigation into Scaling Laws, Invariance, and Implicit Regularization"
_ICLR.cc/2023/Conference — ICLR 2023 poster_

### Official Review · Reviewer_G4Yp · 2022-10-20

**Confidence:** 4
**Correctness:** 3
**Technical Novelty And Significance:** 3
**Empirical Novelty And Significance:** 3
**Recommendation:** 8

**Clarity, Quality, Novelty And Reproducibility:**

Clarity: high; the paper is written in a manner that makes it easy to follow in most parts

Quality: see strengths/weaknesses above

Novelty: medium; that there exists a relationship between augmentation, regularization, and data size is probably well-known, but this paper examines this relationship in an explicit way and adds more puzzle pieces

Reproducibility: high; the authors have provided the code with which the experiments were run

**Strength And Weaknesses:**

**Strengths**
* The topic of the paper is interesting and relevant to ICLR. The concept of an "exchange rate" between augmentations and more data is intuitive and the paper motivates it well.
* The experiments are extensive and cover interesting questions.
* The paper is generally well-written and easy to follow.
* All important design decisions are explained and the used code is included in the supplementary material.
* The detailed analysis of gradient standard deviation during training with and without augmentations is specifically interesting in my opinion.

**Weaknesses**

The extrapolation into regimes where the use of augmentation corresponds to a "negative amount of additional training data" seems a bit strange to me, at least under "normal conditions" like in Fig.1. I think most practitioners would argue that this is not plausible. This seems to correspond to interpolations into an accuracy that approaches the best achievable rates. Also, f_{ref}^{-1} may not be well-defined any more for accuracies that are significantly above the largest observed "reference" accuracy. It looks to me like the extrapolations beyond the max. possible "number of base samples" are more confusing than they help. Further, this may point to potential problems in the functional form of the power law that was used, maybe the used functional form is too simple to yield good results in extrapolation?

This extrapolation effect then seems to also lead to some non-obvious conclusions, e.g. "diverse but inconsistent augmentations [...] are hindrances at scale" where it is not entirely clear to me that the qualification "inconsistent" is obvious from the experiments.

Still regarding extrapolation:
* In Fig3 (left) it seems that for width 4 (and partially for width 8) augmentations *hurt* performance when using more than 100k training samples. This seems quite surprising and would probably warrant an explicit discussion. Or maybe TrivialAug is just not useful in this setting? Similar for Fig.3 (right) and the SwinTransformer.
* In Fig4 (right) it also seems somewhat implausible that for a number of base samples in the thousands, augmentations would be worth a factor of 100 more in training data (even though this is for generalization under distribution shift). Similarly, the graphs seem to indicate that for "horizontal flips", the augmentation is worth "minus half the training data" for ~100k base samples. This is also somewhat implausible to me.

At the start of Sec.3, I had a hard time understanding what exactly "0", "1", etc. mean in Fig.2 and Fig.5. I read Section 3.1.1 several times and I am still not sure how to exactly interpret the experiments. I'd appreciate it if the authors could clarify this.

Similarly, in the second paragraph of Sec 4.2 it was not clear to me how exactly the train and test data are constructed. E.g. often only rotations by multiples of 90 degrees are used for digital images, and the x-range of Fig.7 (left) also only goes up to 4. Does this mean that up to 4 samples were used? But then the test data would be included in the train data for 4 samples per class, which seems not to be the case. I'd appreciate it if the authors could clarify this a bit more.

I had the impression that the conclusions drawn in Sec 4.2 are a bit too strong. The experiment is interesting but it seems ultimately a bit simple and some further analyses (e.g. breakdown of improvements by classes, example successes and failures) are needed. It seems not impossible that for large augmentation ranges, the network could just learn to recognize dominant color distributions in this particular scenario. Can this be ruled out?

**Additional Comments**

Sec. 3.1.2 / Fig. 3 (left) - this seems to point to the fact that augmentation/regularization helps more with increased model capacity. Maybe this is obvious, but I don't think I saw this being discussed in the paper.

I am slightly confused that the authors use a ResNet *18* for CIFAR-like images. In the original paper by He et al. a ResNet *20* was specifically designed for CIFAR-like data, while the *18* version is used for larger ImageNet images. On the other hand, the authors discuss in the Appendix that they used a "usual CIFAR-10 stem" and refer to the "Bag of Tricks" paper. In that paper I did not find any mention of CIFAR, though. I also briefly looked at the code and got the impression that the "18" version seems to use a [2, 2, 2, 2] configuration (which is the classical ImageNet variant as far as I understand). But I may have been confused here or it is considered irrelevant for the experiments here. Could the authors clarify this?

It would also be good if the best achieved accuracies of the networks could be compared to results from the literature. Here, the goal would not be to claim state-of-the-art performance, but it would be good for the reader to see that the experiments are run in a regime that is sufficiently close to other published numbers, such that it remains plausible that the drawn conclusions remain valid in that regime and are not influenced too much by operating far from that point.

A few additional related works that could be relevant (I'm not suggesting all of these need to be included or cited):
* [Bad Global Minima Exist and SGD Can Reach Them](https://arxiv.org/pdf/1906.02613.pdf) has some results on how augmentation is related to "better minima".
* [How to train your ViT? Data, Augmentation, and Regularization in Vision Transformers](https://arxiv.org/pdf/2106.10270.pdf) also discusses the relation between augmentation, regularization, and training data size.
* [A Group-Theoretic Framework for Data Augmentation](https://arxiv.org/pdf/1907.10905.pdf) also seems relevant in this context.

Minor comments/typos (not affecting evaluation):
* p.1 "benefits of augmentations and the number of samples rises": and->as ?
* Fig. 2 (and potentially others): The colors for "1" and "random" are hard to distinguish.
* p.4 "can be attributes simply" -> attributed
* In Fig.6, does "Aug" here mean "only flipping"?
* Sec.4.1: I think another common name for "prediction averaging" is "test-time augmentation", and "orbit selection" is also called "normalization". Maybe it would be useful to mention these, not sure.
* p.7 "We instantiate and E2CNN" -> an
* References: in some reference titles, lowercase is used where it shouldn't, e.g. "lie group", "sgd", "bayesian", "cnns".
* References: in some references, only the ArXiv version is references, while the paper has appeared in a conference or journal, too.

---------------
### Update after authors' response

I would like to thank the authors for their clarifications and the changes made to the paper. I think these have addressed my concerns and I think the quality of the paper has improved overall. I am still a bit worried that the nuances of the results in the extrapolation regime are easy to miss for the reader. (De-emphasizing these in the graphs would be one way to address that.) But overall I think the paper is now above the acceptance threshold. Therefore I am increasing my score.


**Summary Of The Paper:**

The paper analyzes data augmentation in detail for (small) computer vision tasks with (small) networks. In particular, the paper analyzes the following aspects of data augmentation:
* What is the *exchange rate* between augmentations and more data?
* How does augmentation help OOD generalization in comparison to more data?
* How does augmentation relate to invariance in the models?
* Augmentations add a source of randomness to SGD that can help to find "flatter minima".
* A large number of experiments is done, mostly on CIFAR-related datasets, with a variety of architectures and dataset sizes. The exchange rate is based on fitting of power-law curves to the results.


**Summary Of The Review:**

The paper contains several interesting novel ideas and results. However, there are also several issues around extrapolation and interpretation of the results. I therefore think that the paper in its current form is marginally below the acceptance threshold, but I would be happy to increase my score during the discussion period if some of the concerns can be addressed by the authors.

---

> ### Author Response · Authors · 2022-11-16
> **Response to Reviewer G4YP (Part III)**
>
>
> > At the start of Sec.3, I had a hard time understanding what exactly "0", "1", etc. mean in Fig.2 and Fig.5. I read Section 3.1.1 several times and I am still not sure how to exactly interpret the experiments. I'd appreciate it if the authors could clarify this.
>
> We’ve edited the writing in Sec 3.1.1 and Fig.2 and Fig.5. Please let us know if this clarifies the experiments.
>
> > Similarly, in the second paragraph of Sec 4.2 it was not clear to me how exactly the train and test data are constructed. E.g. often only rotations by multiples of 90 degrees are used for digital images, and the x-range of Fig.7 (left) also only goes up to 4. Does this mean that up to 4 samples were used? But then the test data would be included in the train data for 4 samples per class, which seems not to be the case. I'd appreciate it if the authors could clarify this a bit more.
>
> The first task in the controlled experiment in Section 4.2. is to detect which class a rotated sample is from, for the single image experiment, the task can then be described as “given a rotated image, which of these 10 base images generated it”, and is trained with rotations of these 10 images. We then extend this to more base images. Test data for this task consists of novel rotations not seen during training for Fig 7 (left) and novel rotations of unseen images from the same classes in Fig.7 (right).
>
>
> > I had the impression that the conclusions drawn in Sec 4.2 are a bit too strong. The experiment is interesting but it seems ultimately a bit simple and some further analyses (e.g. breakdown of improvements by classes, example successes and failures) are needed. It seems not impossible that for large augmentation ranges, the network could just learn to recognize dominant color distributions in this particular scenario. Can this be ruled out?
>
> The explanation you provide is interesting, and possibly correct, but, for us, the interesting part is that the color distribution would be unaffected by flips and crops. A priori, one might expect that a network trained to recognize rotated images performs equally whether it is trained with flips and crops or without. Yet, we find that these augmentations do provide benefits, even though they are out-of-distribution, here because they are unrelated to the task at hand of recognizing rotated images.
>
> For us, we see the value of this controlled experiment as a stepping stone toward Sec.5., where we show that the interaction of data augmentations with loss landscape of neural networks and increase in stochasticity and flatness can help to explain the question posed by the OOD experiment described in Sec.4.
>
>
> > Sec. 3.1.2 / Fig. 3 (left) - this seems to point to the fact that augmentation/regularization helps more with increased model capacity. Maybe this is obvious, but I don't think I saw this being discussed in the paper.
>
> We now discuss this in modifications of Fig.3 and the takeaway of Sec. 3.1.2.  Thanks for your suggestion!
>
> > I am slightly confused that the authors use a ResNet 18 for CIFAR-like images.
>
> The use of a ResNet-18 model is popular, even on CIFAR-10. As described, the initial stem is modified from the 7x7 strided convolution and maxpool of an ImageNet ResNet to a 3x3 convolution. The rest of the model, i.e. the [2,2,2,2] block structure is kept the same and we use the updated shortcut shape of the  “Bag of Tricks" paper.
> This architecture is substantially larger than a ResNet-20, and is commonly used in applications as the resulting model has a much better accuracy (around 95%) compared to the original, small ResNets described in He 2015 for CIFAR-10, and is also used in analysis as a stand-in for larger models that might be used on larger datasets.
>
> We have included a new Table 12 in the appendix, where we make a direct comparison of CIFAR-10 accuracies of all the architectures with the training routine used in our experiments. We have also added a ResNet20 to our experiments, which we include in Fig.27. We hope this helps to validate that these are reasonable training recipes that return competitive models.
>
> Finally, we’re glad for all minor comments, which we have incorporated into the updated draft, and for remarks on references, which we have reworked and updated.
>
>
> Thank you again for your thoughtful review. We made a significant effort to address your questions, including new results, and would appreciate it if you would consider raising your score in light of our response. Please let us know if you have additional questions we can address.

---

> > ### Comment · Reviewer_G4Yp · 2022-11-17
> > **Thank you for your response**
> >
> > Thank you for your detailed response to the review(s). I have updated my review based on your response and the changes made in the paper.
> >
> > In the spirit of discussion, I would like to respond to some of comments:
> >
> > > negative amount of additional training data
> >
> > It seems the main message in this regime is that augmentation data can hurt if it is inconsistent with the underlying data distribution. This makes intuitive sense. However, cases where this happens in practice should be caught at the evaluation and analysis stage. So entering this regime is maybe not something we would expect in practice, even though I don't doubt that the experiments are consistent with your discussions. Maybe it would be useful to mention this in the paper, because it may be easy for a quick reader to overlook this. And this consideration already occurs in practical situations "intuitively", e.g. if we want to train a model for OCR or for VQA (that includes directional left/right questions), we should not use horizontal image flips as augmentations. But maybe this is getting a bit philosophical.
> >
> > It also felt to me that some of the graphs in the paper use a large portion of the x-axis for the extrapolation regime (which may have high uncertainties) and this could over-emphasize the conclusions that can be drawn from the extrapolations.
> >
> > > functional form of the power law
> >
> > Thank you for running additional experiments on this topic, these are interesting. I'm still wondering whether you have any thoughts on the possibility of f_{ref}^{-1} not being well-defined for accuracies that are significantly above the largest observed "reference" accuracy. So if f_{aug} produces a value that is outside of the range of f_{ref}, Eq. (1) may be problematic? And the interesting values of all the f's are those that lie close to the boundary of the maximum achievable accuracy.
> >
> > > Section 3.1.1
> >
> > I think this is now easier to understand, but let me try to see if I understood correctly: For each bases sample: "0" means that the original sample was used. "1" means that the original sample was thrown away and an augmented sample was used instead (which is "worse", so "1" < "0"). "2" means that for each sample, one augmentation is added. (And similar "n>1" means that "n-1" augmentations were added.) If this iscorrect, then I think my confusion came from the fact that the ordering 0,1,2,4 here has some discontinuity. Further (if my interpretation is correct) maybe a tuple would explain this better as in (n,m) where n is the original sample, m is the number of augmentations, then "0"=(1,0), "1"=(0,1), "2"= (1,1), "4" = (1,3), ... This is just a minor point, though.

---

> > > ### Author Response · Authors · 2022-11-17
> > > **Thank you for the Follow-Up**
> > >
> > > Thank you for the follow-up questions and clarifications. We're happy to discuss further:
> > >
> > > > It seems the main message in this regime is that augmentation data can hurt if it is inconsistent with the underlying data distribution. This makes intuitive sense. However, cases where this happens in practice should be caught at the evaluation and analysis stage. So entering this regime is maybe not something we would expect in practice, even though I don't doubt that the experiments are consistent with your discussions.
> > >
> > > This is certainly one of our messages, but another of the implications that we believe to be interesting in practice is that when applications are scaled from smaller dataset sizes to larger ones, the relative impact of augmentations changes and this might call for a re-evaluation of the choice of the applied augmentations. An application tested and evaluated on one scale might behave suboptimally when deployed with larger dataset sizes.
> > >
> > > At the same time, extrapolating from existing training runs with strong (here diverse and inconsistent) augmentations to future training runs on larger dataset sizes could lead to the wrong conclusion that future improvement through additional data is not very valuable, when in reality future improvement through additional data could be valuable if the augmentation policy is also re-evaluated. To put differently, the accuracy growth of one augmentation strategy could be slower than the growth of another, more consistent policy, or a setup without augmentation.
> > >
> > > > It also felt to me that some of the graphs in the paper use a large portion of the x-axis for the extrapolation regime (which may have high uncertainties) and this could over-emphasize the conclusions that can be drawn from the extrapolations.
> > >
> > > We have reduced all ranges during the previous revision to put more focus on the interpolation regime. Overall, we're a bit ambivalent here. We do want to show the projected development of these graphs in the extrapolation regime to add context and hope that the inherent uncertainty in those regions is appropriately communicated to readers through the change in plotting style going from interpolation to extrapolation regime.
> > >
> > > > I'm still wondering whether you have any thoughts on the possibility of f_{ref}^{-1} not being well-defined for accuracies that are significantly above the largest observed "reference" accuracy.
> > >
> > > I think we missed adressing this in the earlier response. We agree with the possibility that $f_\text{ref}^{-1}$ might not be well-defined for all possible values of $f_\text{aug}$. This happens in Fig.4, where the exchange for TrivialAug is undefined for sample sizes greater than about 3000. This is arguably a bit inconvenient to plot, but we think this is an essential feature of this notion of "exchange rate".
> > >
> > > In the undefined range of this set of experiments in Fig.4, there really is *no exchange* possible: Based on existing experimental data, even an extrapolation to infinite additional in-domain CINIC data would not be enough to increase accuracy on out-of-domain CIFAR-10-C to the same value that is reached with this augmentation.
> > >
> > > We've now added a new Appendix G to include this discussion.
> > >
> > > > Section 3.1.1
> > >
> > > Thank you for your follow-up here as well. Your understanding of "0" and "1" repetitions is correct, however the additional repetitions are also strict replacements of the base data. So,  "0"=(1,0), "1"=(0,1), but actually "2"= (0,2), "4" = (0,4) in your notation. We use replacement in this way to be able to go between one extreme at $(0,1)$ where the dataset is replaced once, and the other extreme at $(0,\infty)$, where augmentations are newly sampled at random every time a sample is evaluated.
> > >
> > > We have further finetuned our description of Section 3.1.1 to better reflect this setup and are we would be glad to further iterate on it.
> > >
> > > Overall, thank you for the additional feedback!

---

> > > > ### Comment · Reviewer_G4Yp · 2022-11-22
> > > > **Thanks again**
> > > >
> > > > I just would like to thank the authors again for another round of well-thought-out responses and changes to the paper.

---

> ### Author Response · Authors · 2022-11-16
> **Response to Reviewer G4YP (Part II)**
>
> > In Fig3 (left) it seems that for width 4 (and partially for width 8) augmentations hurt performance when using more than 100k training samples. This seems quite surprising and would probably warrant an explicit discussion. Or maybe TrivialAug is just not useful in this setting? Similar for Fig.3 (right) and the SwinTransformer.
>
> While these results may appear surprising, they are firmly in the interpolation regime and based on experimentally observed values. We provide code with the supp. material to replicate our experiments.
>
> Looking at raw experimental results,  a thin ResNet of width 4 trained with 192000 samples reaches a validation accuracy of $0.652878 (\pm0.007517)$ without augmentations and $0.613214 (\pm0.010119)$ with the policy of TrivialAug&Flips&Crops evaluated in Fig.3a. Meanwhile, at 12000 samples, training without augmentations reaches $0.410490(\pm 0.011223)$, whereas training with augmentations reaches $0.579227 (\pm 0.004987)$.
>
>
> > In Fig4 (right) it also seems somewhat implausible that for a number of base samples in the thousands, augmentations would be worth a factor of 100 more in training data (even though this is for generalization under distribution shift). Similarly, the graphs seem to indicate that for "horizontal flips", the augmentation is worth "minus half the training data" for ~100k base samples.
>
> These results, again while we agree that they are somewhat surprising, are based on experimental results. For the OOD results in particular though, here we are looking at cases, where no amount of additional base samples is able to replicate gains made through these augmentations. Large factors are observed in the ramp up towards “no amount”, and this is a particularity of the OOD setting.  Put another way, if as the number of train samples tends to infinity, minimizers of the non-augmented loss yield worse OOD accuracy than minimizers of the augmented loss, then there will be sample sizes at which augmented samples are worth vastly more than non-augmented samples.  The same reasoning does not apply to the in distribution case as minimizers of the non-augmented loss will generalize at least as well as minimizers of the augmented loss in the limiting case as the number of training samples tends to infinity.

---

> ### Author Response · Authors · 2022-11-16
> **Response to Reviewer G4Yp (Part I)**
>
> Thank you for your thoughtful feedback. We address each of your questions below including extensive additional experiments and paper edits inspired by your comments.
>
> > The extrapolation into regimes where the use of augmentation corresponds to a "negative amount of additional training data"
>
> Negative amounts of additional training data might sound surprising, but this is a result that also occurs in experiments and not only in our extrapolations, for example in Fig.2, Fig.3, Fig.4.
> Moreover, this result, while potentially surprising, makes mathematical sense.  In the limiting case as more and more data arrives, the loss function saturates around the optimal parameter vector (optimal in terms of test loss), and these minimizers of $\sum_{(x,y)} l(f_\theta(x),y)$ which converge to optimality as the number of training samples increases may not satisfy, for example, horizontal flip invariance.  However, if we train with data augmentation, then we may not find this optimal parameter vector in the limiting case as the minima of $\sum_{(x,y)} l(f_\theta(x),y)+l(f_\theta(T(x)),y)$ may differ from those of the unaugmented loss, where $T$ denotes a data transformation (e.g. horizontal flip).  Intuitively, if there is any degree to which a data augmentation is inconsistent with the underlying data distribution (i.e. $P(x,y) \neq P(T(x),y)$), then with enough data, augmentation will be deleterious and will therefore be worth a negative number of training samples in our setup.
>
> > Further, this may point to potential problems in the functional form of the power law that was used, maybe the used functional form is too simple to yield good results in extrapolation?
>
> We first want to highlight that the functional form used in our study fits experimental results reasonably well with only three parameters, of which only one governs the actual growth behavior.
>
> Prompted by your feedback, we have now also implemented tools from symbolic regression into our analysis and have added a new Appendix D (Fig. 21, 22, 23, 24), where we investigate four alternative functional forms. These new functional forms provide somewhat better fits of the baseline data (maybe unsurprisingly, given that they were discovered on the same data), yet still find similar trends, globally.
>
> > This extrapolation effect then seems to also lead to some non-obvious conclusions, e.g. "diverse but inconsistent augmentations [...] are hindrances at scale" where it is not entirely clear to me that the qualification "inconsistent" is obvious from the experiments.
>
> We include the notion of “inconsistency” following a larger body of previous work describing this concept. In our work, inconsistency between augmented data distribution and true data distribution can best be seen in Fig. 2. With a single replication, which corresponds to replacing every sample in the base dataset with a single augmented sample (and is marked in red), the entire dataset is replaced by one of samples drawn from the augmented data distribution. This is strictly bad for validation accuracy, and this discrepancy between augmented and base distribution indicates their “inconsistency”.
>
> This notion also appears indirectly in Fig.1, where these inconsistent augmentations are exactly those that fall off quicker as more samples are drawn from the real data distribution. Augmentations become a hindrance at scale when their inconsistency with the true data distribution prevents us from finding well-generalizing parameters which may not yield functions that are invariant to the transformation corresponding to that augmentation as discussed above in our response to your question regarding “negative amount…”.

---

### Official Review · Reviewer_WExn · 2022-10-22

**Confidence:** 3
**Correctness:** 4
**Technical Novelty And Significance:** 2
**Empirical Novelty And Significance:** 3
**Recommendation:** 8

**Clarity, Quality, Novelty And Reproducibility:**

This work is clearly written and the message is evident to the reader without any struggle. The experiments are sound and the quality of the work is well done.
They investigate the impact of data augmentation from different views and provide insight into when and where it's more effective. This is a valid and useful idea and it contributes to the field of research.

**Strength And Weaknesses:**

- The idea is nice, focusing on getting a better understanding of the impact of data augmentation is an important research direction. The findings can improve how data augmentation is done in both computer vision and NLP tasks.

- The presentation of "takeaway"s from the analysis of each section is helpful for summarizing the findings.

- The authors mainly focus on computer vision where the image manipulation functions are very straightforward (e.g., flipping an image). I'm wondering how the findings and takeaways of this work generalizes to text augmentation as well. In NLP, by manipulating texts to augment the datasets, we inadvertently introduce noise. That is because [automatic] context manipulation can lead to grammatically or semantically incorrect sentences. What is the authors' insight about their findings in this scenario?

- In figure 3 the authors show that SwinTransformer is less positively impacted by the augmentations in comparison with other architectures. Is that interpreted as how transformers are in general less impacted or is there something about the SwinTransformer architecture in particular?

**Summary Of The Paper:**

This paper investigates the reason why data augmentation is effective in learning models. They look into different hypotheses: how augmentation encourages invariance, variance regularization, providing an additional source of stochasticity. They also look into the scaling laws of augmentation. The findings of this paper shed light on where and when augmentation is most effective in computer vision.

**Summary Of The Review:**

This work is well-motivated, clearly explained, and sufficiently supported by analysis. There are some questions for the authors (see section Strength And Weaknesses), however overall it's a valuable contribution to the field.

---

> ### Author Response · Authors · 2022-11-16
> **Response to Reviewer WExn**
>
> We thank the reviewer for their highly supportive feedback and interesting questions.  Regarding the relationship between our own observations and NLP, we suspect that similar trends may be found in some NLP settings where perhaps aggressive text augmentation is helpful when downstream data is scarce but harmful when data is abundant.  However, a major confounding factor in NLP is that even on tasks with little training data, practitioners are likely to leverage massively pre-trained language models.  Connecting trends in vision and language is an interesting topic for future work.
>
> We also refrain from speculating in our draft about the connection between SwinTransformer and ViTs at large.  Nonetheless, uncovering the inductive biases of transformer architectures and how they manifest in different data augmentation requirements is a topic that interests us and may inform our future research!

---

### Official Review · Reviewer_iXJA · 2022-10-25

**Confidence:** 3
**Correctness:** 3
**Technical Novelty And Significance:** 2
**Empirical Novelty And Significance:** 3
**Recommendation:** 5

**Clarity, Quality, Novelty And Reproducibility:**

While the paper looks quite reproducible, it is unclear how one can fairly compare multiple runs with potentially different levels of training. For example, assuming that the validation metrics are reported from the ones before overfitting, a base model may have peaked at N epoch while an augmented one can peak at M. Without reporting effective flops for each of the runs, it is unclear whether a data augmentation is a worthy trade-off to adding another data sample.

**Strength And Weaknesses:**

While benefits of augmentation are quite well proven empirically, there wasn't too much valuation of one additional data collected. This paper delves into when, how much, and what augmentations can help which is quite practical.

While it lays a good starting point, it mostly deals with empirical finding on a relatively small and simple dataset. Without a theoretical finding, one would need to conduct similar experiments outs to evaluate usefulness of augmentations on a specific dataset, model, and/or task. Given that the shape of the curve can differ by any of those factors, the paper can be seen as only reinforcing intuitions built by others but the conclusion is mostly limited to CIFAR and CINIC dataset.

Also, given that experiments dealt with models with relatively low capacities and very small number of classes, it certainly would have been more interesting to see evaluation done on more complex task where there are more uncertainty in data and where it is difficult to over train (in the paper, model was trained sometimes up to 160 epochs). For example, the benefit of augmentation is less noticeable, at least compared to other charts in the main body of the paper, in the figure 17 and 19 and it certainly is less interesting for SwinTransformer, a model with more capacity.


**Summary Of The Paper:**

The paper tries to quantify the value of augmentation by measuring its effectiveness compared to adding more data. It tried to also examine the effectiveness under different settings such as model size, variants, and out-of-distribution settings. Lastly, it tried to explain why augmentation is effective by measuring gradient standard deviations and noted that it has a regularizing effect because it flattens the gradient.

**Summary Of The Review:**

Augmentation is a technique often employed by ML practitioners for its practical values. The paper tried to quantify trade-off between augmentation vs. one additional labeled data and this framework of trade-off can benefit the practitioner to make the right decision at the right time.

However, despite its very empirical nature, it lacks experiments on different tasks, on datasets of varying size and complexity, and on state-of-the-art models since, at their scales, the augmentation may exhibit very different characteristics. Without them, this paper can only be served as intuition reaffirming data but cannot be used as a rule or a way to generally compute the exchange rate.

---

> ### Author Response · Authors · 2022-11-16
> **Response to Reviewer IXJA (Part II)**
>
> > While the paper looks quite reproducible, it is unclear how one can fairly compare multiple runs with potentially different levels of training. For example, assuming that the validation metrics are reported from the ones before overfitting, a base model may have peaked at N epoch while an augmented one can peak at M. Without reporting effective flops for each of the runs, it is unclear whether a data augmentation is a worthy trade-off to adding another data sample.
>
> In this work, we chose to allocate an equal compute budget to all runs and to thus compare all runs with an exactly equal number of FLOPs.  We agree with your point that it is important to make sure that models trained on more data are not under-trained compared to models trained on few samples and vice versa, and we do exactly this in Appendix E, Figure 25 and 26.  In Figure 25, we see that the number of steps chosen does not influence the trends we observe.  And furthermore, we observe exactly the same trends as previously when we examine peak validation accuracy during training instead of validation accuracy at the end of each run (Fig.26).  Also notable is that the peak validation accuracy is quite similar to the accuracy at the end of training.
>
> These experiments validate that our choice of budget does not affect the implications of Figure 1, and we thank you for pointing out this useful ablation which we have included in our updated draft.
>
> > However, despite its very empirical nature, it lacks experiments on different tasks, on datasets of varying size and complexity, and on state-of-the-art models since, at their scales, the augmentation may exhibit very different characteristics.
>
> We do want to reiterate that our study contains a range of datasets with differing complexity (note also experiments on CIFAR-100, EMNIST, MNIST in the appendix), and that dataset size is a central element of our study and varied in all experiments. The models we consider, provide competitive results for the tasks studied in this work.
>
> Inspired by your comments, we have also put a significant effort into adding an ImageNet experiment, which we believe adds further value to our results.
>
> We hope you can consider raising your score in light of our response.

---

> ### Author Response · Authors · 2022-11-16
> **Response to Reviewer IXJA (Part I)**
>
> Thank you for your helpful feedback. We address each comment below:
>
> > this paper delves into when, how much, and what augmentations can help which is quite practical.
>
> We do want to highlight that we also move further than the analysis of how much augmentations help in Sections 4 and 5 and explore reasons for their qualitative behavior.
>
> >  Without a theoretical finding, one would need to conduct similar experiments outs to evaluate usefulness of augmentations on a specific dataset, model, and/or task. Given that the shape of the curve can differ by any of those factors, the paper can be seen as only reinforcing intuitions built by others but the conclusion is mostly limited to CIFAR and CINIC dataset.
>
> The experimental setup that we construct in this work allows us to provide a detailed analysis based on substantial amounts of experimental data, that would be infeasible for larger datasets. We are confident that there is value in this type of fine-grained analysis.
> We also verify that similar trends hold for other datasets in the appendix, checking MNIST, EMNIST and CIFAR-100.
>
> Prompted by your review, we have now also added results for ImageNet in Appendix F, Fig.28, where we show that the same power laws also fit ImageNet results with several augmentation strategies. We hope you find this to be an interesting addition, that validates that the hypotheses put forward in our work also extend to an even larger dataset with many classes.
>
>
> > Also, given that experiments dealt with models with relatively low capacities and very small number of classes, it certainly would have been more interesting to see evaluation done on more complex task where there are more uncertainty in data and where it is difficult to over train (in the paper, model was trained sometimes up to 160 epochs).
>
> Our experiments are designed with this exact problem in mind.  We try a wide range of low capacity models such as very thin ResNets (which cannot fit the training set) and high capacity models like wide ResNet variants (ones which easily overfit their training data) and find the same trends across this spectrum, see Figure 3a. Further, the ResNet model we consider as baseline is actually a relatively high capacity model (at 11 mio parameters).
>
> We have now included a new Table 12 in the Appendix, where we record the parameter counts for all models investigated in this work to make it straightforward to compare their capacities.
>
>
>
>
> > For example, the benefit of augmentation is less noticeable, at least compared to other charts in the main body of the paper, in the figure 17 and 19 and it certainly is less interesting for SwinTransformer, a model with more capacity.
>
> The SwinTransformer used in this work is a variant adapted to CIFAR-10, ultimately leading to a best-performing architecture with only around 4m parameters. Due to this, the architecture actually has somewhat less capacity than several of the other architectures in Fig. 3b. We agree that benefits of augmentations are less visible in Fig.17 (MNIST), and minor for Fig.19 (EMNIST), but arguably, this is an interesting (though for MNIST not too surprising) outcome by itself.
>
> We’ve since added additional models with larger capacities, such as ResNet-152 in a newly added Fig. 27 in the appendix, showing that observed behaviors are stable across model capacities (as evaluated for capacity via width in Fig.3a), but differ across model families.  Thank you for your point, and these additional experiments and discussions improve our updated paper.

---

### Official Review · Reviewer_VSLP · 2022-10-27

**Confidence:** 4
**Correctness:** 4
**Technical Novelty And Significance:** 2
**Empirical Novelty And Significance:** 4
**Recommendation:** 8

**Clarity, Quality, Novelty And Reproducibility:**

- Clarity: The paper is easy to follow in general (only a few comments; please refer to the previous section). Take-away messages also help to orient the readers.
- Quality: The paper is well-written; the experimental design and result presentation are of high quality.
- Originality: The paper's approach to measure the performance influence of data augmentation is new, to the best of my knowledge.
- Reproducibility: Data sheets and code are attached for reproducibility.


**Strength And Weaknesses:**

Strength
- Data augmentation is an area that has huge practical impact but is short of systematic studies. This paper is an important addition.
- The "exchange rate" concept provides a new tool to measure the effects of data augmentation. I think it has the benefit of being more universal / "normalized" compared with other metrics such as "accuracy improvement", when applied across datasets and model architectures.
- Many aspects / hypotheses are considered; conclusions are drawn using carefully designed experiments.

Weakness
- While I understand that the power-law model is a visually plausible and convenient choice, I don't find it overwhelmingly convincing. For example, in Figure 1 (left) the performance of the baseline seems to also saturate close to 200k base samples, yet the power-law model assumed a linear extrapolation.
- I find the words "repetition"/"repeat" in Sec. 3.1.1 and Figure 2 to be confusing. Literally it seems to mean iterating through the same dataset (with the same order) without generating new data. From the context, however, it seems to represent how many augmented data are generated (measured as multiples of base samples). Could you confirm if the latter understanding is correct? Also, what does "random" mean in the legend of Figure 2?

Other feedback:
- The paper only studies label-preserving data augmentations; conclusions about other data augmentation methods, e.g. mixup, require further research.
- On the benefit of data augmentation for out-of-distribution dataset, it seems to me that links could be drawn from the area of domain adaptation and / or (un)supervised pretraining, etc. Given previous research in those areas, it is not surprising that diverse augmented data may work better than real data from the source domain.

**Summary Of The Paper:**

In this paper, the authors empirically investigate the effects of data augmentation on model performance. They first define a performance metric called "exchange rates", measuring the effects of data augmentation in terms of the size of extra real data required for the same performance. From various perspectives such as consistency and diversity of the augmentation method, base sample size, model capacity and architecture, in-distribution vs. out-of-distribution, they find interesting empirical performance patterns. They further study whether the performance boost can be solely attributed to utilizing invariance in the data distribution. Finally, they link the empirical findings with the implicit regularization and flat minimum literature by measuring the gradient norm and flatness metric of the trained models.

**Summary Of The Review:**

The paper presents a systematic empirical investigation of an important topic in modern machine learning -- data augmentation. The idea of measuring data augmentation performance in "units of data" can be applied and compared across datasets and model architectures, and brings in a new perspective to interpret the performance improvement. The paper is well-organized, with well-controlled experiments and clear presentation of results to support the conclusions of each section. All in all, I think the paper would be a valuable addition to the literature of data augmentation research.

---

> ### Author Response · Authors · 2022-11-16
> **Response to Reviewer VSLP**
>
> Thank you for your extensive and encouraging feedback!
>
> > While I understand that the power-law model is a visually plausible and convenient choice, I don't find it overwhelmingly convincing. For example, in Figure 1 (left) the performance of the baseline seems to also saturate close to 200k base samples, yet the power-law model assumed a linear extrapolation.
>
> Thank you for the question. We do think that the simple power law described in Sec. 3 is a sensible baseline, especially due to its simplicity relative to its overall good fit. We also want to highlight that the performance of the baseline appears linear on the log-log plot in Fig.1 only because the plot is anchored around this curve. Model performance continues to logarithmically improve, yet slightly slower than predicted globally.
>
> Inspired by your feedback, we have now looked at alternative functional forms. To do so, we implement recent symbolic regression solvers and use them to discover new relationships. In a newly added Appendix D, we show four new examples of re-analysis with other functional forms, some of which are not necessarily classical power laws. For all of these results, we ultimately find similar trends to what we observe with the much more generic form in Fig.1.  We keep these results as alternatives in Appendix D for interested readers.
>
> Further, prompted by your comments regarding the notion of “replication” in Sec. 3, we have now updated the draft to make the explanation of Section 3.1.1 and Figure 2 more clear.  Your understanding of “repetition” is indeed correct, but we agree that clarifications are in order nonetheless.  “Random” denotes the case where new augmented views are generated on-the-fly every time that sample is seen, which can equivalently be considered the limiting case of many “repetitions”.
>
> Also, we have included additional references, connecting our OOD conclusions to literature on domain adaptation and generalization.  We also agree that mixing augmentations like mixup and CutMix would also be interesting objects of study for future work.

---

### Author Response · Authors · 2022-11-16
**General Response**

We would like to thank all reviewers for their valuable and extensive feedback and for their many questions. We are glad the reviewers liked the concept of exchange rates, the extensive experiments and detailed analysis presented in our work.
We have updated our draft to address each of the questions that were brought up and to include additional clarifications. We have further included additional appendices containing
* Appendix D: A study of alternative power laws found by symbolic regression. We validate our choice of power law through an extensive study of alternatives discovered via symbolic regression. Our chosen power law remains the most reasonable in terms of complexity versus fit, but we detail other explanations, which might also be of interest to future work in this topic.
* Appendix E: An ablation study of the experimental setup of our experiments. We further provide several experimental results and new visualization where we validate the experimental setup in our work.
* Appendix F: Additional results on ImageNet, showing that power laws as described in Sec. 3 are also appropriate for models trained with and without augmentations on ImageNet with various dataset sizes.


Overall, we are thankful for the commentary provided by the reviewers and find the additional material helpful to strengthen and validate our work. We hope reviewers will consider the timeliness and significance of this work, and the significant effort we put into addressing questions, in their final assessment.

---

### Public Comment · ~Nicholas_Teague1 · 2023-04-24
**Suggestion for additional channel of inquiry**

Hi there, I just had a chance to review your work and wanted to suggest a potential alternate channel of inquiry e.g. for any of your future work. In some of my prior benchmarking of tabular dataset augmentations via noise injections I found that the scale of training corpus prior to augmentation appeared to have material impact on the benefit of augmentations themselves, and such benchmarking partly served to inform a somewhat speculative framing tying together a few loose ends to bridge between theory surrounding the benefits of overparameterization, the double descent phenomenon, and several other related matters - which preprint is available on arxiv as “Geometric Regularization from Overparameterization” (Teague, 2022) [arXiv: 2202.09276]. Long story short, I think the tabular modality has some unique potential for investigating the impacts of data augmentation by way of having avilable reasonable scale training sets that capture the full data generating function (in comarison to vastly more complex domains like images or language) - see appendix C of that paper for instance. Best regards.

---

> ### Author Response · Authors · 2023-04-26
> **Thanks**
>
> Thanks for bringing up this interesting reference, we'll take a closer look!

---

### Decision · Program_Chairs · 2023-01-20

**Decision:**

Accept: poster

**Justification For Why Not Higher Score:**

Some parts of this work seem weak (e.g. about late-training stochasticity), as I mentioned in my meta-review. If the authors indeed improve this in the final version, this paper could be bumped into a spotlight.

**Justification For Why Not Lower Score:**

This is a clear accept, as I wrote: most reviewers vote for a clear accept (8), and one reviewer remained with a marginal reject (5), but did not update their review or engage in discussion following the authors' response, and I believe all the reviewers major concerns were addressed.


**Metareview: Summary, Strengths And Weaknesses:**

This empirical the study examines various questions related to data augmentation in image classification, such as finding the "exchange rate" between augmentations and more data and how it depends on the type of augmentation, model, model width, or if we are using OOD data. Also, it discusses relations to model invariance, gradient stochasticity, and flatness. A considerable amount of experiments is done with a variety of architectures and dataset sizes, so this valuable data. Also, the paper is mostly written well. There are several interesting and novel conclusions, such that the exchange rate can be fitted using a power-law curves to the results, and the shape of this power law is determined by several factors, such as the consistency and diversity of the augmentations.

Most reviewers vote for a clear accept (8), and one reviewer remained with a marginal reject (5), but did not update their review or engage in discussion following the authors' response, and I believe all the reviewers major concerns were addressed.

However, I have a comment, which I hope the authors will address in their final version:
The arguments about stochasticity and its correlation with flatness did not sound very convincing to me. If I understood correctly, the authors suggest that late training stochasticity (due to augmentations) is improving flatness and, therefore, generalization. But I could not find validation accuracy results in this part to show this last part is indeed true. And I doubt this is true since, for example, previous paper showed Batch augmentation is reducing stochasticity but is typically improving generalization.






**Note From Pc:**

if the above contains the word "oral" or "spotlight" please see: "oral" presentation means -> notable-top-5% and "spotlight" means -> notable-top-25%. As stated in our emails, we are disassociating presentation type from AC recommendations

---

> ### Author Response · Authors · 2023-02-28
> **Additional Info**
>
> Thank you for the detailed metareview! We have added additional results correlating flatness with validation accuracy and with empirical exchange rate in a new Appendix H for our camera-ready version.